# GMVALUATOR: SIMILARITY-BASED DATA VALUATION FOR GENERATIVE MODELS

**Jiaxi Yang**[1*†]**, Wenlong Deng**[1,5*]**, Benlin Liu**[2]**, Yangsibo Huang**[3]**, James Zou**[4]**, Xiaoxiao Li**[1,5‡]
[1]University of British Columbia [2]University of Washington [3]Princeton University
[4]Stanford University [5]Vector Institute

## ABSTRACT

Data valuation plays a crucial role in machine learning. Existing data valuation methods, mainly focused on discriminative models, overlook generative models that have gained attention recently. In generative models, data valuation measures the impact of training data on generated datasets. Very few existing attempts at data valuation methods designed for deep generative models either concentrate on specific models or lack robustness in their outcomes. Moreover, efficiency still reveals vulnerable shortcomings. We formulate the data valuation problem in generative models from a similarity matching perspective to bridge the gaps. Specifically, we introduce *Generative Model Valuator* (GMVALUATOR), the first training-free and model-agnostic approach to providing data valuation for image generation tasks. It empowers efficient data valuation through our innovative similarity matching module, calibrates biased contributions by incorporating image quality assessment, and attributes credits to all training samples based on their contributions to the generated samples. Additionally, we introduce four evaluation criteria for assessing data valuation methods in generative models. GMVALUATOR is extensively evaluated on benchmark and high-resolution datasets and various mainstream generative architectures to demonstrate its effectiveness. Our code is available at: https://github.com/ubc-tea/GMValuator.

## 1 INTRODUCTION

As the driving force behind modern AI, particularly deep learning Pei (2020), a substantial volume of data is indispensable for effective machine learning. On one hand, informative data samples relevant to the task at hand play a critical role in the training process. On the other hand, due to data privacy concerns, personal data is safeguarded by various regulations, including the General Data Protection Regulation (GDPR) Regulation (2018), and has become a valuable asset. Consequently, data valuation has garnered significant attention from academic and industrial sectors recently.

The intricate relationship between data and model parameters presents a significant challenge in the contribution measurement of each training sample, thus making data valuation a difficult task. Most existing data valuation studies focus on supervised learning for discriminative models (*e.g.,* classification and regression). These methods can be categorized as *(1) Metric-based methods:* Methods such as *Shapley Value* (SV) and *Banzhaf Index* (BI) Ghorbani & Zou (2019); Wang & Jia (2023) provide the assessment of data value by calculating marginal contribution on performance metrics (*e.g.,* accuracy or loss) through retraining the model[1]. *(2) Influence-based methods:* This line of methods measures data value by evaluating influence on model parameters of data points Jia et al. (2019b); Nohyun et al. (2022). *(3) Data-driven methods:* These techniques avoid retraining by leveraging data characteristics (like data diversity, generalization bound estimation, class-wise distance), though generally necessitate data labels. Xu et al. (2021); Wu et al. (2022); Just et al. (2023).

***Urgent needs for generative models.*** Data valuation in the context of generative models has **NOT** been well-investigated in the current literature. Moreover, there exist significant challenges in

---

*Equal contribution.
†Work was done during visiting at the University of British Columbia
‡Corresponding to: Xiaoxiao Li <xiaoxiao.li@ece.ubc.ca>

[1]Exception for KNN-Shap Jia et al. (2019a)

directly adapting the aforementioned data valuation methods for discriminative models listed to generative models: *Firstly,* the challenge of applying *metric-based methods* arises from lacking robust performance metrics in generative models, in contrast to the existence of commonly used metrics (*e.g.,* accuracy or loss) in discriminative models Betzalel et al. (2022). This could potentially result in inconsistent outcomes when employing various performance metrics Terashita et al. (2021). In addition, the expensive cost of retraining requirements is another obstacle to using *metric-based* methods. *Secondly, influence-based methods* may not perform well on non-convex objective function of generative models Bae et al. (2022). Besides, estimating the influence function in a deep generative model is expensive, as it requires computing (or approximation) inverse Hessian. *Thirdly,* for *data-driven methods*, they mainly focus on supervised learning, which requires knowing data labels to quantify data values.

To the best of our knowledge, limited studies have explored model-dependent data evaluation using influence functions for specific generative models. IF4GAN Terashita et al. (2021) consider multiple evaluation metrics, such as log-likelihood, Inception Score (IS), and Frechet Inception Distance (FID) Heusel et al. (2017) to identify the most responsible training samples for the overall performance of Generative Adversarial Networks (GAN)-based models. However, selecting appropriate metrics is critical, as the results are inconsistent across metrics. VAE-TracIn Kong & Chaudhuri (2021) finds the most significant contributors for generating a particular generated sample for Variational Autoencoders (VAE) using the influence function. However, viable Hessian estimation in influence function calculations incur high computational costs and this method cannot be easily generalized to other generative models.

***Our motivation and goal.*** Considering the challenges posed by the selection of performance metrics and computational efficiency and the need for broader applicability across various generative models, we aim to propose a unified and efficient data valuation method. We expect the unified method to satisfy the following key properties: 1) *Model-agnostic:* the method should be versatile, capable of being employed across diverse generative model architectures and algorithms, regardless of specific design choices; 2) *Computation Efficiency:* the method does not require retraining the model and minimize computational overhead while maintaining a satisfactory level of accuracy and reliability in evaluating the value of data points; 3) *Plausibility:* the method should evaluate the value of data based on its alignment with human prior knowledge on the task, enhancing its credibility and reliability; 4) *Truthfulness:* the method should strive to provide an unbiased and accurate assessment of the value associated with individual data points. In this work, we evaluate the contribution of (good) training data ***given a fixed set of generated data from a certain well-trained generative model***. We refer to the value of training data as the contribution to the given generated data, taking into account the quality of the generated data. To meet the design purposes, we propose a similarity-based data valuation approach [2] for the generative model, called Generative Model Valuator *(GMVALUATOR)*, which is *model-agnostic* and *efficient*.

In summary, our work here provides the following specific novel contributions:

• To the best of our knowledge, GMVALUATOR is the first modal-agnostic and retraining-free data valuation method for image generative models.

• We formulate data valuation for generative models as an efficient similarity-matching problem. We further eliminate the biased contribution measurement by introducing image quality assessment for calibration.

• We propose four evaluation methods to assess the truthfulness of data valuation and evaluate GMVALUATOR on different datasets (including benchmark datasets and high-resolution large-scale datasets) and various deep generative models to verify GMVALUATOR's validity.

***Related Work.*** Different from discriminative models for regression or classification tasks, generative models in various forms (*i.e.,* Variational auto-encoders (VAEs) Rezende et al. (2014), Variational auto-encoders (VAEs) Rezende et al. (2014), Diffusion Model Ho et al. (2020); Rombach et al. (2022)) aim to learn data distribution for generation tasks. Consequently, the approach to data valuation in our work, which focuses on generative models, is distinct and orthogonal to that for discriminative models. The primary limitation of both *metric-based methods* and *influence-based methods* is the expensive computation cost, a drawback that is further magnified in generative models. Conversely, while

---

[2] We detail **why** the similarity-based approach is formed in Sec 2 and explain **how** it is implemented in Sec 3.

data-driven methods are training-free, they predominantly concentrate on supervised learning and data itself Xu et al. (2021); Wu et al. (2022); Just et al. (2023). Apart from the expensive computational cost, the limited existing *influence-based methods* for generative models are model-specific, which can not adapt to the diverse and evolving trends in generative model development Terashita et al. (2021); Kong & Chaudhuri (2021). A more detailed related work is discussed in Sec. C of the appendix.

## 2 MOTIVATION AND PROBLEM FORMULATION

In this section, we present the background and motivation behind our proposed method GMVALUA-TOR, which formulates the data valuation for generative models as a similarity-matching problem.

### 2.1 MOTIVATIONS

Let us denote $X = \{x_1, ..., x_n | x \sim \mathcal{X}\}$ as the training dataset without duplicated data for a generation task, and $n$ is regarded as the size of the dataset. Denote the generated dataset as $\hat{X} = \{\hat{x}_1, ..., \hat{x}_m\}$ by a well trained generative model $G^*$ (*e.g., Generative Adversarial Network* (GAN), *Variational Auto-encoder* (VAE), *Diffusion Model*) on $X$.

The core of these generative models is an intractable probability distribution learning process to train a generator $G$, that maps each $z$ sampled from the noise distribution $\mathcal{Z}$ to a generated sample $\hat{x}$ in estimated distribution by maximizing the likelihood $p_{\mathcal{X}}(x) \approx \int p_G(x|z)p_{\mathcal{Z}}(z)dz$. This can be achieved by minimizing the distance $d(\cdot, \cdot)$ (*i.e.,* , Wasserstein Distance) between the estimated distribution and training data distribution:

$$G^* = \arg\min_G d(G_{z \sim \mathcal{Z}}(z)), \mathcal{X}). \tag{1}$$

Therefore, the data generated by an optimal generator, which closely approximates the training distribution, can be considered as drawing from a subdistribution of $\mathcal{X}$.

### 2.1.1 THEORETICAL JUSTIFICATION

Let $T \subset X$ denote the $K$ contributors found by a data valuator (*e.g.,* GMVALUATOR) for $\hat{X}$, and $S$ is a subset of the training dataset $X$. We focus on the data type with describable attributes (*e.g.,* image data with semantic attributes). We first provide definitions of data and contributors.

**Definition 2.1** *(Data with Describable Attributes.) We assume an image can be characterized by $V$ attributes, and there is a labeling function $f$ mapping $S$, $T$ and $\hat{X}$ to the same attribute space $\mathcal{A} = \{0, 1\}^V$.*

**Definition 2.2** *(Contributors in Generative Models.) Let $S^*$ denotes the real $K$ contributors, defined as follows:*

$$S^* = \arg\min_{S \subset X} d(\mathcal{X}_{(\hat{X}|A)}, \mathcal{X}_{(G(S)|A)}), \tag{2}$$

where $|S^*| = K$ and $K$ is a reasonable number for generative model training. $\mathcal{X}_{(\hat{X}|A)}$ is the data distribution of generated data $\hat{X}$ on attributes $A \in \mathcal{A}$, and $\mathcal{X}_{(G(S)|A)}$ is the distribution of data with attribute $A$ that are generated by the optimal generator trained by contributors $S$. According to the objective of generative models and Eq. equation 2, we have $\mathcal{X}_{(\hat{X}|A)} \sim \mathcal{X}_{(G(S^*)|A)}$. We follow Just et al. (2023) on assumptions and lemmas that will be used to obtain the theorem.

**Assumption 2.3** *Assume that the function $f$ is $\epsilon-$Lipschitz and the loss function $\mathcal{L} : \{0, 1\}^V \times [0, 1]^V \to R^+$ is k-Lipschitz in both inputs and attributes. We have labeling functions that are all bounded by $V$ as $\|f\| \leq V$.*

Then, we introduce the error bound between $T$ and $S^*$.

**Theorem 2.4** *(Bounded Attributes Classification Error on $S^*$ to $T$.) Let $f'_{S^*} : \mu \to \mathcal{A} = \{0, 1\}^V$ be the model trained on the optimal contributor dataset $S^*$. Following Assumption 2.3, if the contributors*

*are corresponding to the given generated data $\hat{X}$, we have:*

$$\mathbb{E}_{x \sim \mu_T} \left[ \mathcal{L} \left( f(x), f'_{S^*}(x) \right) \right] - \mathbb{E}_{x \sim \mu_{S^*}} \left[ \mathcal{L} \left( f(x), f'_{S^*}(x) \right) \right]$$
$$\leq k\epsilon \cdot \left[ d_W (\mathcal{X}_{(T|f)}, \mathcal{X}_{(\hat{X}|f)}) + d_W (\mathcal{X}_{(S^*|f)}, \mathcal{X}_{(\hat{X}|f)}) \right] \tag{3}$$

As shown in the Theorem 2.4, given the optimal contributor $S^*$, $d_W \left( \mathcal{X}_{(S^*|f)}, \mathcal{X}_{(\hat{X}|f)} \right)$ is a deterministic term that approaching zero. Then, approximate $S^*$ can be achieved by reducing the distance term $d_W \left( \mathcal{X}_{(T|f)}, \mathcal{X}_{(\hat{X}|f)} \right)$. Please see Sec. M in the appendix for the proof.

### 2.1.2 EMPERICAL VALIDATION

According to theoretical justification, the generated data should show more similarity to the data samples used for training. In this context, we examine the value of training data being more valuable compared to data that is not used for training, despite originating from a similar distribution. We support this by partitioning a class of CIFAR-10 (the class is plane here) into two non-overlapped subsets, denoted as $X_{v1}$ and $X_{v2}$.[3] Next, we keep $X_{v1}$ as non-training data and use $X_{v2}$ as training data to train a BigGAN Brock et al. (2018) and generate dataset $\hat{X}$. If our assumption holds, the generated data will be more similar to the training data $X_{v2}$. As presented in the T-SNE Van der Maaten & Hinton (2008) plot (Figure 1), the generated data demon-

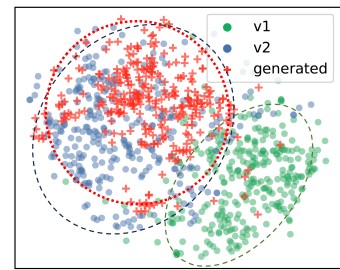

Figure 1: Data distribution for $X_{v1}$, $X_{v2}$ and $\hat{X}$ for CIFAR-10, and $X_{v1}$, $X_{v2}$ are both airplane dataset.

strate a more substantial overlap in the shared feature space with the training dataset $X_{v2}$ than the non-training ones $X_{v1}$.[4] This observation motivates us to address the data valuation for the generative model from a similarity-matching perspective.

## 2.2 PROBLEM FORMULATION AND CHALLENGES

The primary objective of our research is to tackle the issue of data valuation in generative models using *black-box* access. In essence, **given a fixed set of data samples** $\hat{X}$ **with a size of** $m$ **generated from the well-trained deep generative model** $G^*$, our aim is to determine the value $\phi_i(x_i, \hat{X}, G^*)$ associated with each data point $x_i \in X$ for $i \in [n]$ in the *deduplicated* training dataset, which contributes to the generated dataset. We denote the value for training data $x_i$ as $\phi_i$ in the rest of the paper for simplicity.

Following the motivation stated in Sec. 2.1, $\phi_i$ should be a function of the distance between the data points. We denote the distance between training data $x_i$ and generated data $\hat{x}_j$ as $d_{ij}$, for $i \in [n]$ and $j \in [m]$.

**Definition 2.5** *(Primary Contribution Score.)* The contribution score of $x_i$ to $\hat{x}_j$ is denoted as $\mathcal{V}(x_i, \hat{x}_j) \propto d_{ij}^{-1}$, which is inversely proportional to distance since maximizing the log-likelihood of a generative model is equivalent to minimizing the dissimilarity of real and generated data distribution. To link the dissimilarity $-d_{ij}$ to likelihood, we choose $exp(-d_{ij})$ to likelihood

$$\mathcal{V}(x_i, \hat{x}_j) = \frac{\exp(-d_{ij})}{\sum_i \exp(-d_{ij})}. \tag{4}$$

Therefore, an intuitive definition for the value of data sample $x_i$ can be written as below.

---

[3]In practice, we perform T-SNE first to randomly split the non-overlapped samples from the embedding space.

[4]Quantitative statistical testing is provided in Appendix.

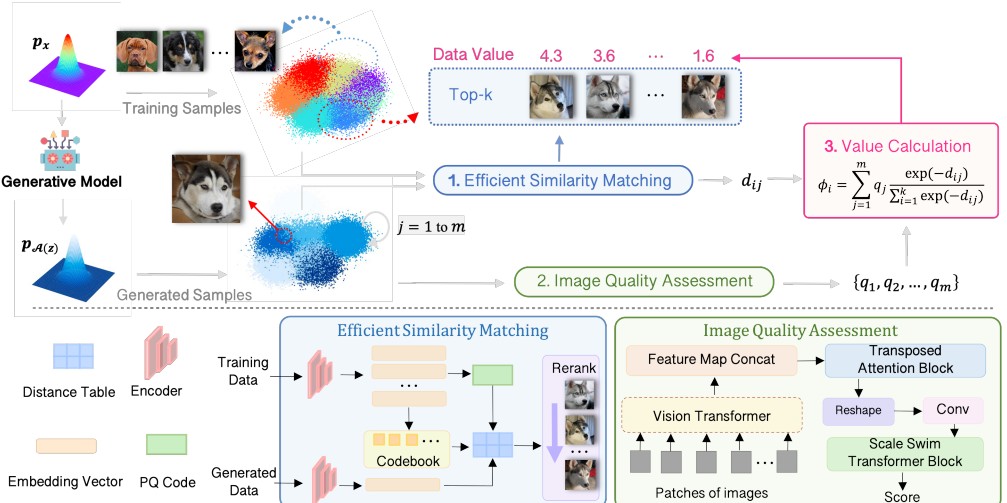

Figure 2: Overview of GMVALUATOR, a unified and training-free data valuation approach for any generative models. GMVALUATOR contains three important modules – *(1) Efficient Similarity Matching (ESM), (2) Image Quality Assessment, and (3)Value Calculation*. Each generated data $\hat{x}_j$ is matched with training data through ESM approach, resulting in the distances with its top $k$ contributors. The normalized contribution score from training sample $x_i$ to $\hat{x}_j$, defined as $\exp(-d_{ij})/\sum_i^k \exp(-d_{ij})$, is adjusted based on the quality of the associated generated samples $q_j$. We compute the data value $\phi_i$ of each training sample $x_i$ by summing its contributions to the generated samples, where it ranks among the top $k$ contributors.

**Definition 2.6** *(Data Value.) The contribution of each training data point $x_i \in S^*$ for the generation of dataset $\hat{X}$ equals to the sum of its contributions to each generated data point $\hat{x}_i$ in $\hat{X}$.*

$$\phi_i = \sum_{j=1}^{m} \mathcal{V}(x_i, \hat{x}_j), x_i \in X, \hat{x}_j \in S^*. \tag{5}$$

The high-value data will achieve a smaller distance to the target distributions, thus, a better approximation. Therefore, data valuation in the above problem formulation contains two steps: *Step 1*, calculating all of the score value $\mathcal{V}(x_i, \hat{x}_j)$; *Step 2*, mapping the contribution from training data to generative data based on the scores $\mathcal{V}$.

However, there are several open questions and challenges in performing the above two steps for calculating Eq. equation 5:

*Challenge 1: Efficiency*. In step 1, considering $n$ training samples and $m$ generated samples, where $\mathcal{O}(C)$ represents the complexity of the selected pair-wise distance calculation, the total complexity of this step amounts to $\mathcal{O}(mnC)$. In practical scenarios with large training datasets (e.g., $n > 10K$), the computation cost becomes prohibitively expensive. Additionally, fitting such a large collection of high-dimensional data for distance calculation can pose significant challenges in system memory.

*Challenge 2: Contribution plausibility*. To ensure that a training data point contributes more if it is similar to high-quality generated data and less if it is similar to low-quality generated data, the contribution scores should be adjusted based on the quality of the generated data.

*Challenge 3: Non-zero scores*. In practical scenarios, the distance between training data and the least similar generated data is not infinite, which may result in false non-zero contribution scores. With a large dataset size, the accumulation of these noisy scores yields biased data valuation.

In light of this, we present a novel and efficient data valuation approach suitable for agnostic generative models, termed as GMVALUATOR as elaborated in Section 3.

## 3 THE PROPOSED DATA VALUATION METHODS

The crucial idea behind GMVALUATOR is to transform the data valuation problem into a similarity-matching problem between generated and training data. The overview of GMVALUATOR is presented in Figure 2. To tackle *challenge 1*, we propose to employ efficient similarity matching (ESM), where

each generated data point can be linked to multiple contributors from the training dataset (Section 3.1). Each generated data firstly is linked with top $k$ contributors via recall phase and then re-ranked by a refined similarity for effectiveness. After that, the image quality of the generated sample is assessed to weigh the valuation (Sec. 3.2). Finally, the value computation function combines both the quality score and the image-space similarity score to measure the value of the training data (Sec. 3.3).

## 3.1 Efficient Similarity Matching

Considering the complexity of calculating $\mathcal{V}(x_i, \hat{x}_j)$, we formulate it as an ESM problem between generative data and training data. Each generated data sample is matched to several training data samples based on their similarity. We denote $\mathcal{P}_j = \{x_1, x_2, ..., x_k\} = f(X, \hat{x}_j)$ as the subset of training data that contains the $k \ll n$ most similar data samples. Here, $f$ represents the similarity-matching strategy, encompassing the *recall* and *re-ranking* phases, which will be introduced subsequently.

**Recall Phase:** The main aim of the *recall* phase is to rapidly identify a subset of training samples that are similar to a generated sample. To achieve this, an initial step involves encoding all original training images and generated samples from the image space $R^C$ to a lower-dimensional embedding space $R^D$ using an pre-trained encoder $f_e$ (such as CLIP Radford et al. (2021)) to reduce computational complexity. Subsequently, the technique of *Product Quantization* (PQ) Jegou et al. (2010) is employed to further decrease the computational burden. Specifically, PQ divides embedding vectors into subvectors and independently quantizes each subvector through $Q$-means clustering. This process generates compact PQ codes that serve as representations of the original vectors. This representation significantly reduces the vector sizes, allowing for efficient estimation of Euclidean Distance between two samples. By incorporating the *recall* method into the similarity matching process, the computational complexity is lowered from $\mathcal{O}(mnC)$ to $\mathcal{O}(mQD)$, where $D \ll C$ and $Q \ll n$. Consequently, generated images can quickly identify their top-$k$ most similar training data samples.

**Re-Ranking Phase:** Following the PQ-based efficient recall process in GMVALUATOR, we further improve the precision of the results by utilizing perceptual similarity Fu et al. (2023) for precision ranking. Once we have extracted perceptual features from the top $k$ recalled training samples, we proceed to calculate the distance for each pair of items. To obtain precise distance measurements based on their perceptual content, we propose to use Learned Perceptual Image Patch Similarity (LPIPS) Zhang et al. (2018) or DreamSim Fu et al. (2023) as the distance measurement $d$ to gain insights into the perceptual dissimilarity between the generated sample and different training samples. These metrics enable us to precisely measure the most significant contributors according to their perceived similarity, more importantly, the obtained distance $d$ will be employed to compute data valuation in Eq. equation 6. We do not use Wasserstein distance as it is more proper to measure dissimilarity between two probability distributions rather than a pair of image instances with semantic characteristics.

## 3.2 Image Quality Assessment

To tackle *challenge 2*, we have to establish a connection between the quality of the generated sample that the training samples contribute to and the contribution score. This is necessary before assigning contribution scores to the ranked training samples for the generated sample. For low-quality generated samples $\hat{x}_{\text{low}}$, we expect their total contribution score from contributors to be lower compared to high-quality samples $\hat{x}_{\text{high}}$, namely $\sum_{i \in [k]} \mathcal{V}(x_i, \hat{x}_{\text{low}}) < \sum_{i \in [k]} \mathcal{V}(x_i, \hat{x}_{\text{high}})$. This motivation arises from the observations in Figure 3, where the first and second rows show generated samples using $z$ from a normal distribution and uniform distribution, respectively. The values denoted in Figure 3 directly employs Eq equation 4, so we have $\sum_{x_i \in \mathcal{P}} \mathcal{V}(x_i, \hat{x}) = 1$ for $\forall \hat{x} \in \hat{X}$. The real samples are noticeably dissimilar to the generated data, yet they are still assigned a high value.

Therefore, we propose to calibrate the contribution scores with the generated data quality. Specifically, we obtain a comprehensive quality score $q_j \in [0, 1]$ for each generated image integrated using MANIQA Yang et al. (2022) model into our evaluation process. A higher $q_j$ indicates better data quality. This score of MANIQA considers various factors such as sharpness, color accuracy, composition, and overall visual appeal. We also provide the necessity of this calibration in the appendix I. Incorporating the image quality evaluation provided by MANIQA allows us to more

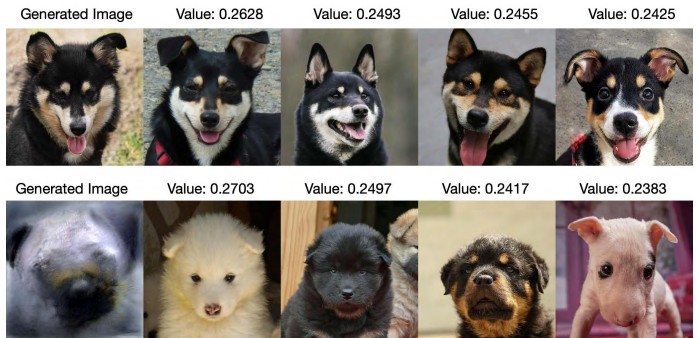

Figure 3: The value without generated image quality calibration for q high-quality image (top row) and a low-quality image (bottom row). Column 1: generated images. Column 2-5:their top 4 contributors.

accurately assess the generated images and take into account their perceptual fidelity and aesthetic qualities.

### 3.3 VALUE CALCULATION

Finally, we utilize the image quality $q$ to calibrate the contribution scores. Notably, during the recall phase of similarity matching, we select the top $k$ contributors $\mathcal{P}_j$ for the generated data $\hat{x}_j$. Consequently, we only consider the contribution score between $\mathcal{P}_j$ and $\hat{x}_j$ by setting $\mathcal{V}(x_i, \hat{x}_j) = 0$ for $x_i \notin \mathcal{P}_j$. This strategy addresses *challenge 3* by assigning zero scores to irrelevant samples, effectively reducing bias and noise in value estimation when dealing with a large $n$. Hence, we define the contribution score of each training data point for a specific synthetic data point as follows:

$$\mathcal{V}(x_i, \hat{x}_j, d_{ij}, q_j) = \begin{cases} q_j \cdot \frac{\exp\left(-d_{ij}\right)}{\sum_{i=1}^{k} \exp\left(-d_{ij}\right)} & x_i \in \mathcal{P}_j \\ 0 & x_i \notin \mathcal{P}_j \end{cases} \quad (6)$$

which is used to give different scores to training data points according to their ranking distance. The final data value is obtained by plugging Eq. 6 into Eq. 5.

## 4 EXPERIMENTS

### 4.1 EXPERIMENT SETUP

**Datasets.** Our experiment settings are listed in Table 1. Particularly, we consider different types of generative models, including GAN based models, $\beta$-VAE Higgins et al. (2016), and diffusion models Ho & Salimans (2022); Wang et al. (2022). The generation tasks are conducted on benchmark datasets (*i.e.,* MNIST LeCun et al. (1998) and CIFAR Krizhevsky et al. (2009)), face recognition dataset (*i.e.,* CelebA Liu et al. (2018)), high-resolution image dataset with size $512 \times 512$, and $1024 \times 1024$ (*i.e.,* AFHQ Choi et al. (2020), FFHQ Karras et al. (2019)), the large-scale dataset with 1,000 classes and 14,197,122 images (*i.e.,* ImageNet Deng et al. (2009)), and text-to-image dataset (*i.e.,* Naruto Cervenka (2022)).

Table 1: Evaluation Setup.

|        | Dataset | Network |
|--------|---------|---------|
| **C1** | MNIST, CIFAR-10 | GAN, Diffusion Models, $\beta$-VAE |
|        | ImageNet | Masked Diffusion Transformer Gao et al. (2023) |
| **C2** | CelebA | Diffusion-StyleGAN |
|        | AFHQ, FFHQ | StyleGAN |
|        | Naruto | Stable Diffusion |
| **C3** | MNIST | DCGAN |
| **C4** | **Hardware Evironment** | |
| GPU | One RTX 3080 (10GB) GPU | |
| CPU | 12 vCPU Intel(R) Xeon(R), Platinum 8255C CPU @ 2.50GHz | |

**Baselines.** Note that GMVALUATOR is a model-agnostic method. To showcase its efficacy, we compare it with two baseline methods: VAE-TracIn Kong & Chaudhuri (2021) and IF4GAN Terashita et al. (2021). VAE-TracIn identifies the most significant contributors for a generated sample, which is the same as the similarity matching process of GMVALUATOR (Sec. 3.1). Thus, we compare GMVALUATOR with VAE-TracIn in Sec. 4.3 and Sec. 4.4. IF4GAN finds the high-valued training samples for generative model training, which we regard it as the baseline method in section 4.5. We also examine our approach using DreamSim, LPIPS and $l_2$-distance in the re-ranking phase, referred to as GMValuator (DreamSim), GMValuator (LPIPS) and GMValuator ($l_2$-distance) respectively, while the GMValuator without the re-ranking phase is referred to as GMValuator (No-Rerank).

**Embedding Appoarch.** In this experiment, we utilize CLIP as an example choice for a pre-trained encoder $f_e$ in the recall phase of efficient similarity matching. To demonstrate the generalizability of GMVALUATOR, we also do experiments using more embedding approaches in Sec. G of the appendix.

## 4.2 METRICS

Given the absence of a definitive benchmark to evaluate data valuation methods in the context of generative models, we propose to evaluate the truthfulness of data valuation outcomes through the examination of four criteria (C):
- **C1: Identical Class Test.** Following the concept of an identical class Hanawa et al. (2020), we posit that the most significant contributors among the training samples should belong to the same class as the generated sample produced by a well-trained generative model, which is also used for evaluation in VAE-TrancIn Kong & Chaudhuri (2021).
- **C2: Identical Attributes Test.** Following **C1**, we extend the concept of identical class test and consider the attributes of samples as the ground truth to examine the most significant contributors by our approach on datasets without class labels. We examine the overlap level of attributes (*i.e.,* the number of identical attributes) between the most significant training samples with the generated sample.
- **C3: Out of Distribution Detection.** Under the assumption, the out of distribution training data samples (*e.g.,* noisy samples) can be identified as low-contribution training samples for the generated dataset. To evaluate the performance of data valuation approaches, we consider the contribution level of noisy data samples as the performance metric across various approaches.
- **C4: Efficiency.** As an efficient training-free approach, GMVALUATOR should measure the data value in a limited time and be much more efficient than the previous work while obtaining truthful results.

## 4.3 IDENTICAL CLASS TEST (C1)

***Methodology.*** In this subsection, we follow setup of VAE-TracIn Kong & Chaudhuri (2021) by training separate $\beta$-VAE models on MNIST Krizhevsky et al. (2009), CIFAR LeCun et al. (1998). We then attribute the most significant contributors by VAE-TracIn and our approach over generated samples. By the concept of identical class test, we expect a perfect data valuation approach should indicate training data in the same subclass of the generated sample $\hat{x}_j$ contribute more to $\hat{x}_j$. We examine GMVALUATOR (DreamSim), GMVALUATOR (LPIPS), GMValuator ($l_2$-distance) and GMVALUATOR (No-Rerank) separately. In addition to using VAE and comparing it with the baseline method, we also perform experiments using masked diffusion transformer on the large-scale dataset (ImageNet) and other various generative models (in the appendix) to demonstrate the model-agnostic property of GMVALUATOR.

***Results.*** For a given generated data, we examine the class(es) of is top $k$ contributors in the training data. We count the number of training $Q$ samples in the top $k$ contributors, which have the identical class as the generated data. The identical class ratio $\rho = Q/k$. We report the averaged $\rho$ over the generated datasets (the data size $m$=100) on different choices of $k$ in Table 2. GMVALUATOR has the most contributors that belong to the same class with the generated sample among on MNIST and CIFAR-10, while GMVALUATOR (DreamSim) has the best performance. The results for CIFAR-10 using VAE-TracIn are extremely bad. This could be attributed to underfitting, as the authors mentioned in their study Kong & Chaudhuri (2021), training a good $\beta$-VAE model on CIFAR-10 is challenging. In addition, GMVALUATOR (DreamSim) shows a significant advantage on ImageNet since it considers semantic characteristics while $l_2$-distance or LPIPS destroy the performance, while the experiments conducted on ImageNet using baseline method (VAE-TracIn) is impractical due to

its unacceptable computational overhead. In the appendix, further analysis and explanation for this phenomenon are discussed and the results from various generative models are presented as well.

Table 2: Performance comparison of Identical Class Test (C1).

| Top-$k$ | MNIST (%) | | | CIFAR-10 (%) | | | ImageNet (%) | | |
|---|---|---|---|---|---|---|---|---|---|
| | $k$=30 | $k$=50 | $k$=100 | $k$=30 | $k$=50 | $k$=100 | $k$=30 | $k$=50 | $k$=100 |
| VAE-TracIn | 72.00 | 71.11 | 68.58 | 6.28 | 3.77 | 1.88 | - | - | - |
| GMValuator (No-Rerank) | 86.41 | 85.13 | 85.95 | 72.66 | 72.25 | 70.71 | 56.18 | 53.60 | 48.17 |
| GMValuator ($l_2$-distance) | 87.76 | 86.95 | 85.92 | 72.66 | 72.25 | 70.71 | 42.44 | 40.51 | 39.00 |
| GMValuator (LPIPS) | 88.78 | 88.19 | 86.69 | 72.60 | 71.75 | 70.84 | 50.30 | 47.51 | 45.13 |
| GMValuator (DreamSim) | 88.78 | 88.05 | 86.84 | 77.94 | 76.41 | 73.47 | 77.27 | 70.90 | 58.89 |

## 4.4 IDENTICAL ATTRIBUTES TEST (C2)

***Methodology.*** In extension to **C1**, we focus on certain image attributes instead of class labels as the ground truth. We posit that the most significant contributors to a generated sample should share similar attributes with it. We train Diffusion-StyleGAN Wang et al. (2022) on CelebA, and leverage our approach to identify the most significant contributors in training samples and use some attributes including hat, gender, and eyeglasses as the ground truth, to check the correctness of our approach for identifying the most significant contributors for a generated sample. We also examine this on AFHQ, FFHQ dataset by StyleGAN Choi et al. (2020), and text-to-image dataset Naruto Cervenka (2022) by stable diffusion model Rombach et al. (2022).

***Results.*** The results in Table 3 indicate that GMVALUATOR can find the most significant contributors with the same attributes to the generated sample. Besides, using DreamSim in the similarity re-ranking phase obtains the best performance over these three attributes. We also visualize some visible results in Figure 4 on CelebA, and AFHQ, FFHQ in the appendix. As we can see, the most significant contributors have similar attributes (*e.g.,* skin color, hair) to the generated sample. Similarly, the visualized results on AFHQ and FFHQ shown in the appendix demonstrate that the top $k$ contributors have similar attributes to the generated sample such as the fur color of cats or dogs in AFHQ or hair color in FFHQ. In addition, the results on the text-to-image dataset Naruto Cervenka (2022) corroborate the same conclusion, as shown in the appendix N.

Table 3: Performance of Identical Attributes Test (C2) of some attributes including Hat, Gender, and Eyeglasses on CelebA.

| Attribute (%) | Hat | | | Gender | | | Eyeglasses | | |
|---|---|---|---|---|---|---|---|---|---|
| Top K contributors: | $k$=5 | $k$=10 | $k$=15 | $k$=5 | $k$=10 | $k$=15 | $k$=5 | $k$=10 | $k$=15 |
| GMValuator (No-Rerank) | 92.52 | 92.12 | 91.92 | 75.15 | 75.25 | 75.82 | 91.52 | 91.52 | 91.45 |
| GMValuator ($l_2$-distance) | 90.91 | 89.90 | 89.83 | 63.64 | 61.01 | 60.00 | 94.95 | 94.65 | 93.80 |
| GMValuator (LPIPS) | 96.77 | 96.57 | 96.90 | 97.98 | 97.48 | 97.37 | 94.95 | 94.65 | 93.80 |
| GMValuator (DreamSim) | 97.78 | 97.07 | 96.90 | 99.19 | 98.99 | 98.79 | 96.77 | 96.26 | 95.96 |

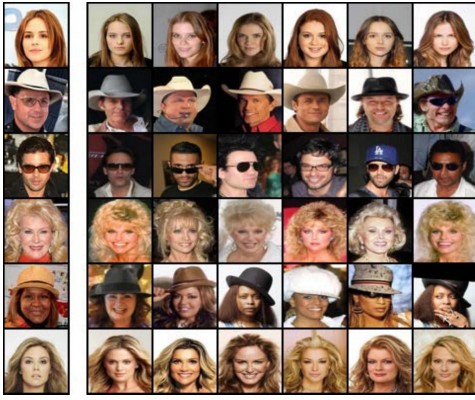

Figure 4: Visualization of Identical Attributes Test on CelebA. Left: generated samples. Right: top $k$ contributors.

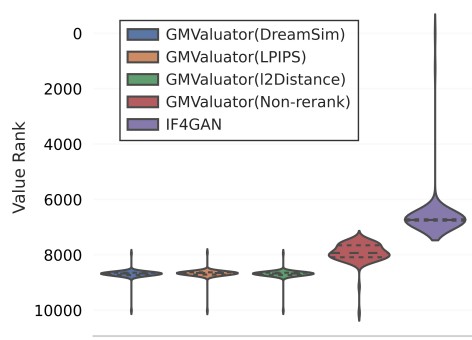

Figure 5: The *y-axis* represents the ranking of values from high to low, with the top being the highest value and the bottom being the lowest value. The *x-axis* represents the index of each noisy data.

### 4.5 Out of Distribution Detection (C3)

***Methodology*** In this experiment, we introduce 100 noisy images into the MNIST dataset to create a new corrupted training dataset and each noisy image is generated by adding Gaussian noise to the clean data. These contaminated samples can diminish the performance of generative models, and as a result, they are anticipated to possess lower values compared to the rest of the training dataset. Our objective is to assess the performance of both GMVALUATOR and its baseline method, IF4GAN Terashita et al. (2021), by analyzing the value rankings of noisy samples. An effective approach should place noisy data in lower positions within the value ranking. We conducted this evaluation using 10,000 generated images and referred to the study Terashita et al. (2021) for the baseline method (IF4GAN).

***Results.*** The results depicted in Figure 5 demonstrate that the values of all noisy training samples, as calculated by *GMValuator*, are lower compared to the values calculated by IF4GAN. This observation suggests that the performance of GMVALUATOR is significantly better than that of IF4GAN. Besides, the performance of GMVALUATOR with the re-ranking phase is better than GMVALUATOR (No-rerank).

### 4.6 Efficiency (C4)

***Methodology.*** We thoroughly evaluate and compare the efficiency of our approach against existing data valuation methods for generative models discussed above: VAE-TracIn and IF4GAN. Specifically, we measure the attribute time for one generated sample on an average of 100 test samples using both VAE-TracIn and our proposed approach on MNIST and CIFAR-10. Furthermore, under the same setting in **C3**, we assess the data valuation time when utilizing IF4GAN and our approaches on noise MNIST which we mentioned in **C3**. In addition, we analyze our computational efficiency as the data scale grows in the appendix O.

***Results.*** The average time taken to attribute the most significant contributors for one generated sample is calculated and compared with the baseline method (VAE-TracIn), as demonstrated in Table 4. Our approaches are all greatly more efficient than the baseline methods on both datasets. When it comes to data valuation time for GAN on **C3**, GMVALUATOR (No-rerank) and GMVALUATOR (LPIPS) are significantly better than IF4GAN. The aforementioned phenomenon is attributed to the costly nature of the Hessian estimation process, despite the utilization of certain acceleration methods. It is noticeable that GMVALUATOR (DreamSim) is lightly time-consuming when compared to IF4GAN. This lead to a trade-off problem due to the GMVALUATOR using DreamSim in the re-ranking phase obtained the better performance from **C1** to **C3**.

Overall, through the above experiments, GMVALUATOR outperforms than baseline methods, while GMVALUATOR (DreamSim) obtains the best performance. This is due to the fact that DreamSim captures mid-level similarities in image semantic content and layout compared to LPIPS. And the selection of different GMVALUATORs can be combined with the consideration of their efficiency. Furthermore, the results shown in the appendix O also demonstrate that our method maintains consistent computational efficiency even as the data scale increases and presents its scalability on large-scale datasets.

Table 4: Efficiency Comparison

| ***Attribute for VAE*** | VAE-TracIn | GMValuator | | | |
| --- | --- | --- | --- | --- | --- |
| | | No-Rerank | $l_2$-distance | LPIPS | DreamSim |
| MNIST | 47.945 | 0.250 | 0.339 | 0.477 | 1.709 |
| CIFAR-10 | 66.178 | 0.755 | 1.226 | 2.412 | 15.491 |
| ***Data Valuation for GAN*** | IF4GAN | GMValuator | | | |
| | | No-Rerank | $l_2$-distance | LPIPS | DreamSim |
| Noise MNIST | 14,543 | 2,137 | 3,388 | 4,771 | 17,086 |

## 5 Conclusion

To measure the contribution of each training data sample, we propose an efficient approach, GM-VALUATOR, for generative model. As far as we are aware, there is no prior model-agnostic and training-free data valuation approach for generative models util GMVALUATOR. Our approach is based on efficient similarity matching, and it enables us to calculate the final value of each training data point, aligning with plausible assumptions. The proposed method is validated through a series of comprehensive experiments to showcase its truthfulness and efficacy on four criteria. In the future, we will validate the proposed methods on other data modalities.

## 6 ACKNOWLEDGEMENT

This work is supported in part by Natural Science and Engineering Research Council of Canada (NSERC), Canada CIFAR AI Chairs program, Canada Research Chair program, MITACS-CIFAR Catalyst Grant Program, the Digital Research Alliance of Canada, Institute of Information & communications Technology Planning & Evaluation (IITP) grant funded by the Korea government (MSIT).

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

CONTENTS

**Roadmap of Appendix.** In this appendix, we provide a concise summary of key notations and Algorithm 1, detailing the pipeline of GMValuator in Sec. A and Sec. B. A more comprehensive review of related work is also provided in Sec. C. Additionally, Sec. D contains detailed statistical analysis for the results depicted in Figure 1. Our main text focuses on experiments conducted on benchmark datasets such as MNIST and CIFAR-10, along with a large-scale dataset, ImageNet, to evaluate the most significant contributors found by GMVALUATOR are in the same class as the generated sample (**C1**). In Sec. E, we extend the evaluation of GMVALUATOR to other generative models for **C1**. Additionally, we expand our evaluation using **C2** by considering the ground truth of multiple combined attributes. We operate under the assumption that the most significant contributors to a generated sample should exhibit similar attributes in the images. For a more thorough validation of the effectiveness of GMVALUATOR, high-resolution datasets like AFHQ and FFHQ are also employed for **C2**. The appendix includes additional experimental insights such as ablation studies, alternative embedding approaches, alternative distance metrics, calibration and further experimental details in Sec. F, H, I and J, respectively. The potential applications, limitations and future directions are also provided in Sec. K and Sec. L. Lastly, we present the proof of our theorem and ensure the reproducibility of our experiments.

Table 5: Some important notations in Sec. 2

| Notations | Description |
|---|---|
| $\mu$ | the distribution of the subset of the training data |
| $\mu_T$ | a subset of the contributors |
| $G$ | generative model |
| $G^*$ | well-trained generative model |
| $\mathcal{Z}$ | noise distribution |
| z | a latent sample in the latent space |
| $d(\cdot, \cdot)$ | distance function |
| $X, x$ | training dataset, training sample |
| $\hat{X}, \hat{x}$ | generated dataset, generated sample |
| $S$ | a subset of the training data |
| $S^*$ | the real $K$ contributors |
| $K$ | $K = |S^*|$ |
| $T$ | $K$ contributors found by a data valuator |
| $\mathcal{A}$ | attribute space |
| $\mathcal{X}$ | data distribution |
| $\mathcal{L}$ | loss function |
| $f, f'_{S^*}$ | labeling function, model trained on $S^*$ |

## A  NOTATION

To make motivation and problem formulation clear, we list some significant notations from Sec. 2 in Table 5.

## B  ALGORITHM

To better introduce GMVALUTOR, the pseudocode is shown in Algorithm 1.

## C  RELATED WORK

### C.1  DATA VALUATION

There are three lines of methods on data valuation: *metric-based methods*, *influence-based methods* and *data-driven methods*. In terms of *metric-based methods*, the commonly-used approach is to calculate its *marginal contribution* (MC) based on performance metrics (*e.g.,* accuracy, loss). As the basic method depending on performance metrics for data valuation, LOO (Leave-One-Out) Cook (1977) is used to evaluate the value of the training sample by observational change

---

**Algorithm 1** GMVALUATOR

---

**Input**: Training dataset $X = \{x_i\}_{i=1}^n$, a well-trained
model $G^*$, random distribution $\mathcal{Z}$.
**Output**: Generated dataset $\hat{X}$, the value of training data
points $\Phi = \{\phi_1, \phi_2, ..., \phi_n\}$

 1: $\hat{X} = \{\hat{x}_j\}_{j=1}^m \leftarrow G^*(z_j)$, for $z_j \in \mathcal{Z}$    // *Generate the synthetic dataset*
 2: **for** $\hat{x}_j$ in $\hat{X}$ **do**
 3:     // *Matching process (see Sec. 3.1)*
 4:     $\mathcal{P}_j = f(X, \hat{x}_j)$    // *Including two phases*
 5:     **for** $x_i$ in $\mathcal{P}_j$ **do**
 6:         $d_{ij} \leftarrow \text{DreamSim}(x_i, \hat{x}_j)$ or others
 7:     **end for**
 8:     $q_j = MANIQA(\hat{x}_j)$    // *Image Quality Assessment (see Sec. 3.2)*
 9:     Calculate score $\mathcal{V}(x_i, \hat{x}_j, d_{ij}, q_j)$ using Eq. equation 6 // *Contribution Score Calculation (see Sec. 3.3)*
10: **end for**
11: // *Calculation of data value and return the result* $\Phi$
12: **for** $x_i$ in $X$ **do**
13:     Calculate $x_i$'s value $\phi_i$ using Eq.equation 5
14: **end for**
15: **return** $\Phi = \{\phi_1, \phi_2, ..., \phi_n\}$

---

of model performance when leaving out that data point from the training dataset. To overcome inaccuracy and strict desirability of LOO, SV Ghorbani & Zou (2019) and BI Wang & Jia (2023) originated from *Cooperative Game Theory* are widely used to measure the contribution of data Jia et al. (2019b); Ghorbani et al. (2020). Considering the joining sequence of each training data point, SV needs to calculate the marginal performance of all possible subsets in which the time complexity is exponential. Despite the introduction of techniques such as Monte-Carlo and gradient-based methods, as well as others proposed in the literature, approximating data significance value (SV) is computationally expensive and it typically requires retraining Ghorbani & Zou (2019); Jia et al. (2019a). The computational cost and need for unconventional performance metrics present difficulties in adapting the methods to generative models. As for *influence-based methods*, they evaluate the influence of data points on model parameters by computing the inverse Hessian for data valuation Jia et al. (2019a); Richardson et al. (2019); Saunshi et al. (2022). Due to the high computational cost, some approximation methods have also been proposed Pruthi et al. (2020). In addition, the use of influence function for data valuation is not limited to discriminative models, but can also be applied to specific generative models such as GAN and VAE Terashita et al. (2021); Kong & Chaudhuri (2021). When it comes to data-driven methods, most of them are training-free methods that focus on the data itself Xu et al. (2021); Wu et al. (2022); Just et al. (2023). Additionally, the Information-Theoretic Measures method Boopathy et al. (2023) evaluates the inherent generalization difficulty of tasks and shows potential for adaptation to data valuation. However, it is not applicable to our specific setting due to its reliance on performance metrics, training requirements, and a focus on dataset-level trends.

## C.2 GENERATIVE MODEL

Generative models are a type of unsupervised learning that can learn data distributions. Recently, there has been significant interest in combining generative models with neural networks to create *Deep Generative Models*, which are particularly useful for complex, high-dimensional data distributions. They can approximate the likelihood of each observation and generate new synthetic data by incorporating variations. Variational auto-encoders (VAEs) Rezende et al. (2014) optimize the log-likelihood of data by maximizing the evidence lower bound (ELBO), while generative adversarial networks (GANs) Goodfellow et al. (2020); Karras et al. (2020) involves a generator and discriminator that compete with each other, resulting in strong image generation. Recently proposed diffusion models Ho et al. (2020); Rombach et al. (2022) add Gaussian noise to training data and learn to recover the original data. These models use variational inference and have a fixed procedure with a high-dimensional latent space.

Table 6: The statistic test of data values of $X_{v1}$ versus $X_{v2}$ using different generative models. $X_{v1}$ is supposed to have higher value than $X_{v2}$, given the generated data.

$$H_0: \phi(D_i, S, \mu_i) \geq \phi(D_j, S, \mu_i)$$
$$H_1: \phi(X_i, S, \mu_i) < \phi(X_j, S, \mu_i), i \in X_{v1}, j \in X_{v2}$$

|  | **BigGAN** | **Classifier-free Guidance Diffusion** |
|---|---|---|
| Average value (v1) | 0.319654 | 0.030434 |
| Average value (v2) | 1.632352 | 0.369565 |
| P-value | $6.937027 \times 10^{-68}$ | $8.053195 \times 10^{-55}$ |
| T-statistic | 17.924512 | 15.947860 |
| Significance level | 0.01 | 0.01 |
| Result | p-value less than 0.01, reject $H_0$, value of v2 less than v1 averagely | p-value less than 0.01, reject $H_0$, value of v2 less than v1 averagely |

## D    STATISTICAL RESULTS FOR FIGURE 1

It is evident by visualization in Figure 1 that the data points in $X_{v2}$ (used for training) are more overlapped with generated data than data points in $X_{v1}$ (not used for training). We perform statistic testing on data values obtained by GMVALUATOR, to examine if data points $X_{v2}$ (used for training) have significantly higher values than those of the data points in $X_{v1}$ (not used for training).

To this end, we use a t-test with the null hypothesis that data values in $X_{v1}$ should not be smaller than those of $X_{v12}$. We compute $p$-value, which is the probability of getting a difference as large as we observed, or larger, under the null hypothesis. If the $p$-value is very low, we reject the null hypothesis and consider our approach, GMVALUATOR, to be verified with a high level of confidence $(1-p)$. Typically, a $p$-value smaller than significance level 0.01 is used as a threshold for rejecting the null hypothesis. Table 6 showcases the outcomes of $X_{v1}$ and $X_{v2}$ in CIFAR-10 with $p \ll 0.01$ for both BigGAN and diffusion model, indicating that the data points in $X_{v2}$ have significantly more value than those in $X_{v1}$. Consequently, these findings align with the presumption that the trained dataset $X_{v2}$ has a higher value than the untrained dataset $X_{v1}$ and verify our approach.

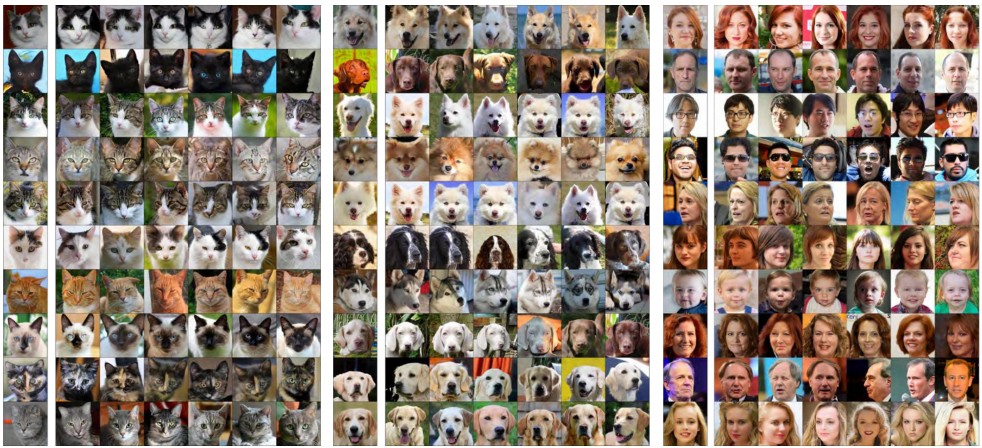

Figure 6: Visualization of Identical Attributes Test on AFHQ and FFHQ. The results shown in the first and second subfigures on the left are conducted on AFHQ-Cat and AFHQ-Dog, respectively. The subfigure on the right presents the results of FFHQ. In each subfigure, the generated samples are on the left, and the top $k$ contributors in the training dataset are on the right.

Table 7: Performance comparison of Identical Class Test.

| MNIST | | | |
| --- | --- | --- | --- |
| GAN (%) | $k$=30 | $k$=50 | $k$=100 |
| GMValuator (No-Rerank) | 96.27 | 96.26 | 95.86 |
| GMValuator ($l_2$-distance) | 97.73 | 97.58 | 96.03 |
| GMValuator (LPIPS) | 97.77 | 97.72 | 97.38 |
| GMValuator (DreamSim) | 97.43 | 97.44 | 97.40 |
| Diffusion (%) | $k$=30 | $k$=50 | $k$=100 |
| GMValuator (No-Rerank) | 92.40 | 91.82 | 91.26 |
| GMValuator ($l_2$-distance) | 92.90 | 92.66 | 91.88 |
| GMValuator (LPIPS) | 93.73 | 97.72 | 92.42 |
| GMValuator (DreamSim) | 93.90 | 93.44 | 92.55 |
| CIFAR-10 | | | |
| BigGAN (%) | $k$=30 | $k$=50 | $k$=100 |
| GMValuator (No-Rerank) | 64.70 | 63.80 | 62.14 |
| GMValuator ($l_2$-distance) | 64.70 | 63.80 | 62.14 |
| GMValuator (LPIPS) | 63.67 | 62.80 | 61.51 |
| GMValuator (DreamSim) | 70.33 | 68.74 | 65.18 |
| Class-free Guidance Diffusion (%) | $k$=30 | $k$=50 | $k$=100 |
| GMValuator (No-Rerank) | 72.67 | 72.00 | 71.00 |
| GMValuator ($l_2$-distance) | 72.67 | 72.00 | 71.00 |
| GMValuator (LPIPS) | 72.53 | 72.28 | 71.06 |
| GMValuator (DreamSim) | 79.37 | 78.08 | 74.61 |

# E  ADDITIONAL RESULTS ON C1 AND C2

## E.1  (C1) IDENTICAL CLASS TEST ON OTHER GENERATIVE MODELS

We have presented Identical Class Test (C1) on $\beta$-VAE and MNIST LeCun et al. (1998), CIFAR-10 Krizhevsky et al. (2009) in Sec. 4.3 in our main context. Since GMVALUATOR is model-agnostic, we further validate our method of **C1** on other generative models.

Here, we conduct the experiments using a GAN and a Diffusion Model on MNIST. The architectural details of the used generative models are described in Sec. J.3 in the appendix. We also conduct the experiment on BigGAN Brock et al. (2018) and Class-free Guidance Diffusion Ho & Salimans (2022) with CIFAR-10. We used the same number of generated samples $m = 100$ as the experiments presented in Sec. 4.

Following the similar settings in Sec. 4.3 (**C1**), we examine the class(es) of is top $k$ contributors for a given generated data in the training data. We calculate the number of training samples, denoted as $Q$, from the top $k$ contributors that have the same class as the generated data. The identical class ratio, denoted as $\rho$, is calculated as $\rho = Q/k$. We report the average value of $\rho$ across the generated datasets for different choices of $k$ in Table 7. GMVALUATOR (DreamSim) has the highest ratio of contributors that belong to the same class as the generated sample among most of the models evaluated on MNIST and CIFAR-10 datasets for different values of $k$. And the ratio improves as the value of $k$ decreases, which is consistent with the top $k$ assumption and validates our method.

## E.2  (C2) IDENTICAL ATTRIBUTES TEST ON CELEBA

We extend **C1** to focus on the attributes present in the images, treating them as ground truth rather than class labels, as discussed in Sec. 4.4. Our experiments now incorporate multiple attributes simultaneously, rather than just a single attribute, to verify the performance of GMVALUATOR. For instance, when evaluating a generated image with both a hat and eyeglasses, the most significant contributors identified should also include both of these attributes. Table 8 showcases some outcomes of identical attributes test when regarding multiple combined attributes as ground truth, which validate the effectiveness of our methods.

Table 8: Performance of Identical Attributes Test (C2) of multiple combined attributes.

| Top K contributors: | k=5 | k=10 | k=15 |
|---|---|---|---|
| **Attribute**: *Eyeglasses & Gender* (%) | | | |
| GMValuator (No-Rerank) | 50.10 | 48.48 | 48.42 |
| GMValuator ($l_2$-distance) | 59.19 | 56.67 | 56.23 |
| GMValuator (LPIPS) | 78.18 | 74.24 | 73.87 |
| GMValuator (DreamSim) | 92.53 | 90.61 | 90.30 |
| **Attribute**: *Eyeglasses & Hat* (%) | | | |
| GMValuator (No-Rerank) | 78.79 | 78.38 | 85.86 |
| GMValuator ($l_2$-distance) | 84.44 | 82.93 | 83.23 |
| GMValuator (LPIPS) | 86.87 | 86.16 | 86.33 |
| GMValuator (DreamSim) | 89.49 | 87.58 | 87.41 |
| **Attribute**: *Gender & Hat* (%) | | | |
| GMValuator (No-Rerank) | 58.59 | 57.47 | 57.71 |
| GMValuator ($l_2$-distance) | 61.62 | 60.51 | 60.40 |
| GMValuator (LPIPS) | 63.84 | 63.23 | 62.96 |
| GMValuator (DreamSim) | 65.25 | 64.44 | 64.18 |

### E.3 (C2) IDENTICAL ATTRIBUTES TEST ON OTHER DATASETS

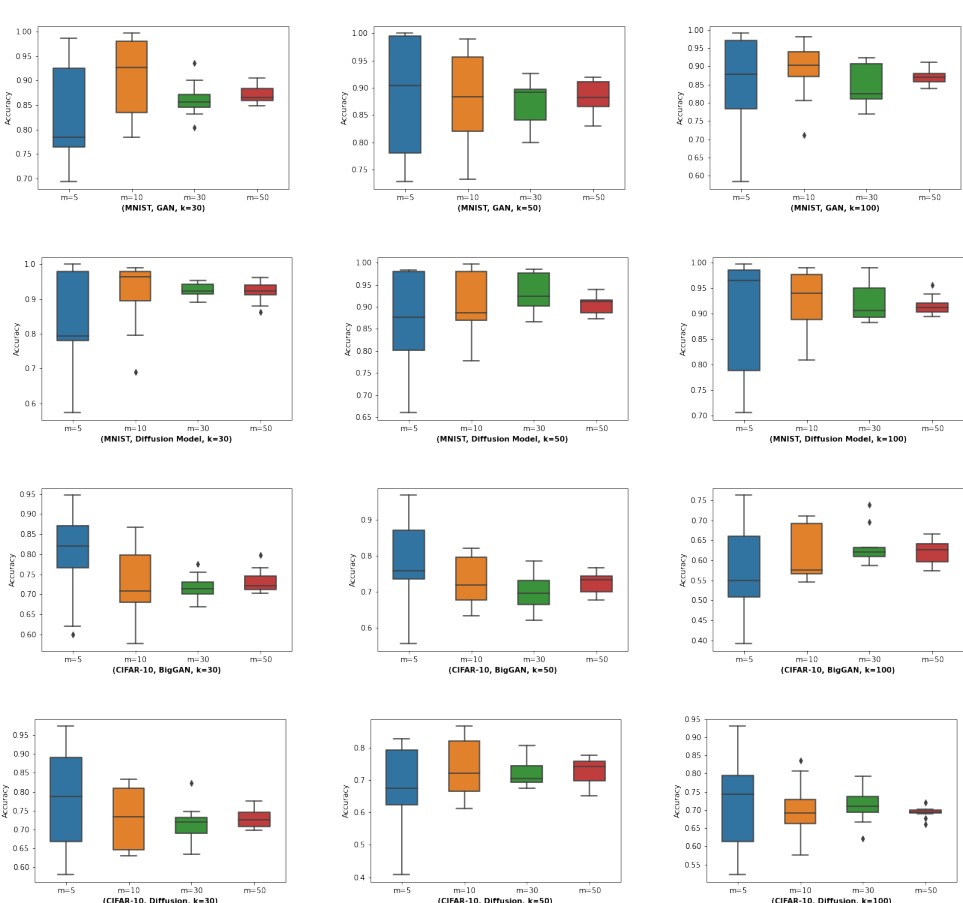

Figure 7: The change of $\rho$ with the different number of generated samples $m$ on MNIST and CIFAR-10 by diverse generative models.

To further validate GMVALUATOR, we also conducted experiments on high-resolution datasets: AFHQ Choi et al. (2020) and FFHQ Karras et al. (2019), following the same settings in Sec 4.4. The results are shown in Figure 6, which demonstrates the effectiveness of our methods in **C2**. The results show that the top $k$ contributors have similar attributes with the generated sample such as fur color of cats or dogs in AFHQ. For the experiment conducted on FFHQ, human faces attributes of the most significant contributors are also similar to the attributes in generated images.

## F  DIFFERENT GENERATED DATA SIZES

Since our value function $\phi_i$ (Eq. equation 5) for training data $x_i$ is computed by averaging over generated samples, it is expected that the sensitivity of $\phi_i$ is connected to the size $m$ of the investigated generated sample. To explore the influence of generated data size $m$ on the utilization of GMVALUATOR, we perform sensitivity testing on MNIST, CIFAR10 using generative models GAN Goodfellow et al. (2020), and Diffusion models Dhariwal & Nichol (2021) as depicted below. The dataset, model and used $k$ are denoted under each subfigure of Figure 7. First, we generate a varying number of samples from the same class. Specifically, we consider four different sample sizes, denoted by $m$, which are given by 1, 10, 30, and 50. Next, we evaluate the GMVALUATOR using parameter **C1**, and this evaluation is performed for each of the aforementioned values of $m$. Subsequently, we conduct the experiment 10 times using GMVALUATOR (No-Rerank), each time with different $m$-sized generated data samples from the same class. The results are presented as the mean and standard deviation ($\rho$) for accuracy, taken over these 10 runs. The results shown in Figure 7 imply that varying $m$ does not yield notable differences in mean accuracy and increasing the number of generated samples $m$ leads to more stable and consistent results.

## G  ALTERNATIVE EMBEDDING APPROACHES

In the step of efficient similarity matching, the embedding $f_e$ is exclusively used in the recall phase for retaining $n$ samples with non-minimal contributions. Apart from using CLIP for embedding, we also investigate the influence for the results when using other embedding methods. We present results in Table 9 using more embedding methods, showing similar performance with CLIP with "Rerank". The reason for this results is that we implement re-ranking that *relies on the image space rather than the embedding space*, which allows us to derive accurate top $k$ contributors from $n$ samples $n \gg k$. Thus, the embedding could tolerate some noise in the recall phase.

Table 9: Comparison of different embedding methods.

| MNIST (%) | No-Rerank | $l_2$-distance | LPIPS | DreamSim |
|-----------|-----------|----------------|-------|----------|
| CLIP      | 86.41     | 87.76          | 88.78 | 88.78    |
| Alexnet   | 79.77     | 84.82          | 86.51 | 88.72    |
| Densenet  | 80.47     | 86.77          | 87.97 | 89.35    |

## H  ALTERNATIVE DISTANCE METRIC

We suggest utilizing Learned Perceptual Image Patch Similarity (LPIPS) Zhang et al. (2018) or DreamSim Fu et al. (2023) as the distance metric $d$ during the re-ranking phase. This enables to understand the perceptual dissimilarity between generated and real data points. These metrics are applied in the image embedding space derived from pre-trained models. Alternatively, distance measurement can be based on pixel space, such as employing $l2$-distance. Notably, we find that distances calculated in the input pixel space yield comparable outcomes as shown in Table 7. This implies that our method GMVALUATOR could be flexible to the different choices of distance metrics and the selection can depend on data prior.

## I  NECESSITY FOR CALIBRATION

For challenges 2 and 3, we utilize image quality assessment and establish a non-zero scores rule for calibration in GMVALUATOR. To better understand the impact and necessity of calibration,

we compare the distribution of the top 1, 2, and 3 contributors' scores with (w) and without (w/o) calibration conducted on MNIST in Figure 8. It is evident that scores without calibration are generally higher than with calibration. We also extend C3 and perform an ablation study in Figure 9 on scenarios where the generated samples include low-quality outputs. The results show that the OOD (out-of-distribution) training samples' value rank by GMValuator without (w/o) calibration is smaller, indicating significant bias and poor performance.

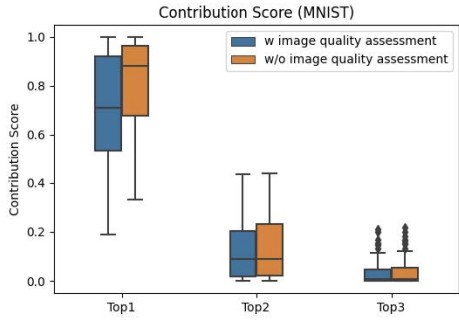

Figure 8: Sore Distribution

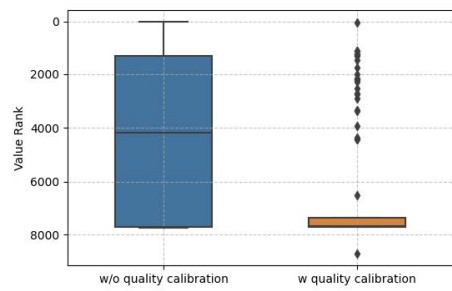

Figure 9: Value Rank

## J  ADDITIONAL EXPERIMENTAL DETAILS

### J.1  JUSTIFICATION OF EXPERIMENT SETUP

We provide a detailed justification in Table 10 for our experiment setup from **C1** to **C4**.

- **C1.** In Identical Class Test, the baseline method is VAE-TracIn Kong & Chaudhuri (2021), which can find the most influenced instances in the training dataset. Since VAE-TracIn is the model-specific method, we only need to compare it with GMVALUATOR when VAE model is used. Besides, all the datasets used in **C1** should have the class labels. Considering the computational demands detailed in VAE-TracIn Kong & Chaudhuri (2021), the runtime complexity of VAE-TracIn correlates with the number of network parameters and the size of the dataset. Therefore, our analysis primarily utilizes simpler benchmark datasets such as MNIST and CIFAR-10 for comparative evaluations with VAE-TracIn.

- **C2.** In extending **C1**, we use image attributes to supplant the concept of class for datasets lacking class labels. For datasets with attribute labels, quantified experiments are feasible, as demonstrated in Table 3 and Table 8. For datasets without attribute labels, we employ visualized experiments.

- **C3.** IF4GAN, a model-specific method for GAN, serves as the baseline for measuring data value. Adhering to the settings outlined in Terashita et al. (2021) and considering the impractical computational costs, we conduct experiments on the same dataset (MNIST) in Terashita et al. (2021) using DCGAN.

Table 10: Justification of Experiment Setup. The selection of datasets and models was based on three critical factors that guided the process. Firstly, attribute labels were required to evaluate C2 effectively. Secondly, benchmark datasets were meticulously chosen to ensure a fair comparison with baselines while also taking into account computational costs (C3 and C4). Finally, the selected generative models are powerful enough to generate good-quality data for the datasets.

|    | Dataset | Model | Baseline | Label Requirements |
|----|---------|-------|----------|--------------------|
| **C1** | MNIST, CIFAR-10 | VAE | VAE-TracIn Kong & Chaudhuri (2021) | Class labels |
|    |  | Diffusion, GAN | - | Class labels |
|    | ImageNet | Masked Diffusion Transformer Gao et al. (2023) | - | Class labels |
| **C2** | CelebA | Diffusion-StyleGAN | - | Attribute labels |
|    | AFHQ, FFHQ | StyleGAN | - | - |
| **C3** | MNIST | DCGAN | IF4GAN Terashita et al. (2021) | Class labels |
| **C4** | MNIST, CIFAR-10 | VAE | VAE-TracIn Kong & Chaudhuri (2021) | Class labels |
|    | MNIST | DCGAN | IF4GAN Terashita et al. (2021) | Class labels |

- **C4.** We present a comparison of the efficiency of GMVALUATOR against baseline methods. In accordance with the settings used for VAE-TracIN and IF4GAN in Kong & Chaudhuri (2021), we report the efficiency results for **C1** on the MNIST and CIFAR-10 datasets, and for **C3** on MNIST.

## J.2 DATASETS

We conduct the generation tasks in the experiments on benchmark datasets (*i.e.,* MNIST LeCun et al. (1998) and CIFAR Krizhevsky et al. (2009)), face recognition dataset (*i.e.,* CelebA Liu et al. (2018)), high-resolution image dataset AFHQ Choi et al. (2020) and FFHQ Karras et al. (2019), large-scale image dataset ImageNet Deng et al. (2009).

*MNIST.* The MNIST dataset consists of a collection of grayscale images of handwritten digits (0-9) with a resolution of 28x28 pixels. The dataset contains 60,000 training images and 10,000 testing images.

*CIFAR-10.* CIFAR-10 dataset consists of 60,000 color images in 10 different classes, with 6,000 images per class. The classes include objects such as airplanes, cars, birds, cats, deer, dogs, frogs, horses, ships, and trucks. Each image in the CIFAR-10 dataset has a resolution of 32x32 pixels.

*CelebA.* The CelebA dataset is a widely used face recognition and attribute analysis dataset, which contains a large collection of celebrity images with various facial attributes and annotations. The dataset consists of more than 200,000 celebrity images, with each image labeled with 40 binary attribute annotations such as gender, age, facial hair, and presence of eyeglasses.

*AFHQ.* The AFHQ dataset is a high-resolution image dataset that focuses on animal faces (*e.g.,* dogs, cat), and it consists of high-resolution images with $512 \times 512$ pixels.

*FFHQ.* The FFHQ dataset is a high-resolution face dataset that contains high-quality images (1024x1024 pixels) of human faces.

*ImageNet.* ImageNet is a large-scale image dataset, which contains over 14 million images and is categorized into more than 20,000 classes.

## J.3 ARCHITECTURE OF GENERATIVE MODELS

In our experiments, we leverage different generative models in the class of GAN, VAE and diffusion models. We utilize $\beta$-VAE for both MNIST and CIFAR-10 datasets while a simple GAN is conducted on MNIST. BigGAN and $\beta$-VAE are also conducted on CIFAR-10. We list the architecture details for these generative models from Table 11 to Table 13. StyleGAN is used for high-resolution datasets AFHQ and FFHQ. CelebA uses Diffusion-StyleGAN Wang et al. (2022), for which we use the exact architecture in their open-sourced code. In addition, Masked Diffusion Transformer, as introduced by Gao et al. Gao et al. (2023), is applied to the ImageNet.

Table 11: The architecture of GAN for MNIST.

| Generator | Discriminator |
|---|---|
| FC(100, 8192), BN(32), ReLU | Conv2D(1, 128, 4, 2, 1), BN(128), LeakyReLU |
| Conv2D(128, 64, 4, 2, 1), BN(64), ReLU | FC(8192, 1024), BN(1024), LeakyReLU |

Table 12: The architecture of BigGAN.

| | |
|---|---|
| **Input** | 28×28×1 (MNIST) & 32×32×3 (CIFAR-10). |
| **Encoder** | Conv 32×4×4 (stride 2), 32×4×4 (stride 2), 64×4×4 (stride 2), 64×4×4 (stride 2), FC 256. ReLU activation. |
| **Latents** | 32 |
| **Decoder** | Deconv reverse of encoder. ReLu acitvation. Gaussian. |

Table 13: The architecture of $\beta$-VAE.

| $\beta$-**VAE** | |
| :---: | :---: |
| **Generator** | **Discriminator** |
| $z \in \mathbb{R}^{120} \sim \mathcal{N}(0, I)$ 
 $\text{Embed}(y) \in \mathbb{R}^{32}$ | RGB image $x \in \mathbb{R}^{32 \times 32 \times 3}$ |
| Linear $(20 + 128) \to 4 \times 4 \times 16ch$ | ResBlock down $ch \to 2ch$ |
| ResBlock up $16ch \to 16ch$ | Non-Local Block $(64 \times 64)$ |
| ResBlock up $16ch \to 8ch$ | ResBlock down $2ch \to 4ch$ |
| ResBlock up $8ch \to 4ch$ | ResBlock down $4ch \to 8ch$ |
| ResBlock up $4ch \to 2ch$ | ResBlock down $8ch \to 16ch$ |
| Non-Local Block $(16 \times 16)$ | ResBlock down $16ch \to 16ch$ |
| ResBlock up $2ch \to ch$ | ResBlock $16ch \to 16ch$ |
| BN, ReLU, $3 \times 3$ Conv ch $\to 3$ | ReLU, Global sum pooling |
| Tanh | Embed $(y) \cdot h + ($ linear $\to 1)$ |

## K  DISCUSSION ON THE POSSIBLE APPLICATIONS

The application of data valuation within generative models offers a wide range of opportunities. A potential use case is to quantify privacy risks associated with generative model training using specific datasets, since the matching mechanism GMVALUATOR can help re-identify the training samples given the generated data. By doing so, organizations and individuals will be able to audit the usage of their data more effectively and make informed decisions regarding its use.

Another promising application is material pricing and finding in content creation. For example, when training generative models for various purposes, such as content recommendation or personalized advertising, data evaluation can be used to measure the value of reference content.

In addition, GMVALUATOR can play an important role in the development of ensuring the responsibility of using synthetic data in safe-sensitive fields, such as healthcare or finance. By assessing the value of the data used in generative model training, researchers can ensure that the generated data are robust and reliable.

Last but not least, the applications of GMVALUATOR can promote the recognition of intellectual property rights. Determining the value of the intellectual property being generated by generative models is critical. By evaluating the data employed in training generative models, we can develop a more comprehensive understanding of copyright that may emerge from the generative models. In essence, such insights can help advance licensing agreements for the utilization of the generative model and its outputs.

## L  CURRENT LIMITATION AND FUTURE DIRECTIONS

The limitation of this work is that it only measures data value for vision-related generative models and conducts experiments exclusively within the field of computer vision. However, this does not mean that GMVALUATOR cannot be easily adapted to Natural Language Processing (NLP) fields, given its core idea of similarity matching. In the future, we should extend GMVALUATOR to NLP and assess the data value for language-related generative models, such as large language models (LLMs).

## M  OMITTED PROOFS

We follow Just et al. (2023) to prove the theorem. Firstly, we give several assumptions that will be used in later proof.

**Assumption M.1** *Following Assumption 2.3, given a distance function $d(\cdot, \cdot)$ between , we defined the coupling between $\mathcal{X}_{(T|f)}$ and $\mathcal{X}_{(S^*|f)}$ as $\pi^*$:*

$$\pi^* := \underset{\pi \in \Pi(\mathcal{X}_{(T|f)}, \mathcal{X}_{(S^*|f)})}{\arg\inf} \mathbb{E}_{(x_T, x_{S^*}) \sim \pi} d(x_T, x_{S^*}) \tag{7}$$

It is easy to see that all joint distributions defined above are couplings between the corresponding distribution pairs. Then, following Just et al. (2023) we prove the main Theorem.

**Theorem M.2** *(Restated of Theorem 2.4.) Let $f'_{S^*} : \mu \to \mathcal{A} = \{0,1\}^V$ be the model trained on the optimal contributor dataset $S^*$. Following Assumption 2.3, if the contributors are corresponding to the given generated data $\hat{X}$, we have:*

$$\mathbb{E}_{x \sim \mu_T} \left[ \mathcal{L} \left( f(x), f'_{S^*}(x) \right) \right] - \mathbb{E}_{x \sim \mu_{S^*}} \left[ \mathcal{L} \left( f(x), f'_{S^*}(x) \right) \right]$$
$$\leq k\epsilon \cdot \left[ d_W(\mathcal{X}_{(T|f)}, \mathcal{X}_{(\hat{X}|f)}) + d_W(\mathcal{X}_{(S^*|f)}, \mathcal{X}_{(\hat{X}|f)}) \right] \tag{8}$$

**Proof M.3**

$$\mathbb{E}_{x \sim \mu_T} \left[ \mathcal{L} \left( f(x), f'_{S^*}(x) \right) \right] = \mathbb{E}_{x \sim \mu_T} \left[ \mathcal{L} \left( f(x), f'_{S^*}(x) \right) \right]$$
$$- \mathbb{E}_{x \sim \mu_{S^*}} \left[ \mathcal{L} \left( f(x), f'_{S^*}(x) \right) \right] + \mathbb{E}_{x \sim \mu_{S^*}} \left[ \mathcal{L} \left( f(x), f'_{S^*}(x) \right) \right]$$
$$\leq \mathbb{E}_{x \sim \mu_{S^*}} \left[ \mathcal{L} \left( f(x), f'_{S^*}(x) \right) \right]$$
$$+ \left| \mathbb{E}_{x \sim \mu_{S^*}} \left[ \mathcal{L} \left( f(x), f'_{S^*}(x) \right) \right] - \mathbb{E}_{x \sim \mu_T} \left[ \mathcal{L} \left( f(x), f'_{S^*}(x) \right) \right] \right| \quad (9)$$

*We bound $\left| \mathbb{E}_{x \sim \mu_{S^*}} \left[ \mathcal{L} \left( f(x), f'_{S^*}(x) \right) \right] - \mathbb{E}_{x \sim \mu_T} \left[ \mathcal{L} \left( f(x), f'_{S^*}(x) \right) \right] \right|$ as follows:*

$$\left| \mathbb{E}_{x \sim \mu_{S^*}} \left[ \mathcal{L} \left( f(x), f'_{S^*}(x) \right) \right] - \mathbb{E}_{x \sim \mu_T} \left[ \mathcal{L} \left( f(x), f'_{S^*}(x) \right) \right] \right|$$
$$= \left| \int_{\mathcal{X}^2} \mathcal{L} \left( f(x_{S^*}), f'_{S^*}(x_S) \right) - \mathcal{L} \left( f(x_T), f'_{S^*}(x_T) \right) d\pi^*(x_T, x_{S^*}) \right|$$
$$= | \int_{\mathcal{X}^2} \mathcal{L} \left( f(x_{S^*}), f'_{S^*}(x_{S^*}) \right)$$
$$- \mathcal{L} \left( f(x_{S^*}), f'_{S^*}(x_T) \right) + \mathcal{L} \left( f(x_{S^*}), f'_{S^*}(x_T) \right) - \mathcal{L} \left( f(x_T), f'_{S^*}(x_T) \right) d\pi^*(x_T, x_{S^*}) |$$
$$\leq \int_{\mathcal{X}^2} \left| \mathcal{L} \left( f(x_{S^*}), f'_{S^*}(x_{S^*}) \right) - \int_{\mathcal{X}^2} \mathcal{L} \left( f(x_{S^*}), f'_{S^*}(x_T) \right) \right| d\pi^*(x_T, x_{S^*})$$
$$+ \int_{\mathcal{X}^2} \left| \mathcal{L} \left( f(x_{S^*}), f'_{S^*}(x_T) \right) - \int_{\mathcal{X}^2} \mathcal{L} \left( f(x_T), f'_{S^*}(x_T) \right) \right| d\pi^*(x_T, x_{S^*}) \quad (10)$$

*Then due to $k$-Lipschitzness of $\mathcal{L}$ and $\epsilon$-Lipschitzness of $f$, we can obtain:*

$$RHS\ of\ Eq.equation\ 10 \leq k \int_{\mathcal{X}^2} ||f'_{S^*}(x_{S^*}) - f'_{S^*}(x_T)|| d\pi^*(x_T, x_{S^*})$$
$$+ k \int_{\mathcal{X}^2} ||f(x_{S^*}) - f(x_T)|| d\pi^*(x_T, x_{S^*})$$
$$\leq k\epsilon \int_{\mathcal{X}^2} 2d(x_T, x_{S^*}) d\pi^*(x_T, x_{S^*})$$
$$= k\epsilon d_W(\mathcal{X}_{(T|f)}, \mathcal{X}_{(S^*|f)}),$$

*where the last step is due to the definition of 1-Wasserstein distance. Then, according to the triangle inequality of Wasserstein distance Peyré et al. (2019), we can obtain:*

$$d_W(\mathcal{X}_{(T|f)}, \mathcal{X}_{(S^*|f)}) \leq d_W(\mathcal{X}_{(T|f)}, \mathcal{X}_{(\hat{X}|f)}) + d_W(\mathcal{X}_{(\hat{X}|f)}, \mathcal{X}_{(S^*|f)}) \tag{11}$$

*Combining Eq. equation 9 and Eq. equation 11 we finished the proof and obtained the Theorem 2.4. By reducing the distance term $d_W\left(\mathcal{X}_{(T|f)}, \mathcal{X}_{(\hat{X}|f)}\right)$, we have $\mathcal{X}_{(T|f)} \to \mathcal{X}_{(S^*|f)}$. As a result, the expected distance*

$$\mathbb{E}_{(S^* \sim \mathcal{X}_{(S^*|f)}, T \sim \mathcal{X}_{(T|f)})} \min_{\pi \in \Pi(T, S^*)} \mathbb{E}_{(x_T, x_{S^*}) \sim \pi} d(x_T, x_{S^*}) \to 0,$$

*with randomly sampling $S^*$ and $T$ with $K$ elements.*

# N  EXPERIMENT ON TEXT TO IMAGE GENERATION MODEL (STABLE DIFFUSION)

In this section, we conducted experiments using a stable diffusion model Rombach et al. (2022) fine-tuned with LoRA on a Naruto dataset Cervenka (2022).

***Identical Attributes Test (C2).*** To evaluate the effectiveness of GMVALUATOR on a text-to-image dataset, we use the attributes in the images as ground truth and employ the Identical Attributes Test (C2) to assess its performance. The results presented in Table 14 show that the top $k$ contributors share similar attributes with the generated sample, such as gender, hair, and hat, demonstrating the effectiveness of our approach in evaluating data contributions.

Table 14: Performance of Identical Attributes Test (C2) of some attributes including Gender, Hat, and Hair on Naruto.

| Attribute (%) | Gender | | | Hat | | | Hair | | |
|---|---|---|---|---|---|---|---|---|---|
| Top K contributors: | $k=3$ | $k=4$ | $k=5$ | $k=3$ | $k=4$ | $k=5$ | $k=3$ | $k=4$ | $k=5$ |
| GMValuator (l2_distance) | 94.33 | 93.50 | 92.80 | 86.30 | 85.50 | 84.40 | 88.00 | 87.75 | 88.40 |
| GMValuator (LPIPS) | 98.67 | 98.50 | 98.80 | 87.00 | 86.25 | 85.40 | 96.67 | 96.25 | 96.00 |
| GMValuator (DreamSim) | 99.00 | 98.50 | 98.80 | 86.33 | 86.75 | 85.80 | 98.30 | 97.75 | 96.00 |

***Efficiency.*** To evaluate computational efficiency, we report the time taken to identify significant contributors for a single generated sample in Table 15. The results emphasize the effectiveness and efficiency of our approach for text-image models such as Stable Diffusion. The experiments were conducted in an environment equipped with an RTX A6000 (48GB) GPU and a 12-vCPU Intel(R) Xeon(R) Platinum 8255C CPU @ 2.50GHz.

Table 15: Efficiency using Stable Diffusion on Naruto.

| Method | GMValuator (l2_distance) | GMValuator (LPIPS) | GMValuator (DreamSim) |
|---|---|---|---|
| Time (s) | 1.27888 | 3.09580 | 6.28144 |

# O  EFFICIENCY WITH INCREASED DATA SIZE

As a training-free and efficient data valuation approach for generative models, GMVALUATOR is also highly scalable to large-scale and complex datasets. To evaluate this, we assess the efficiency of GMVALUATOR on datasets of varying sizes, illustrating the trend in computational efficiency as the data size increases. In Fig.10, we present the running time versus the dataset size for GMVALUATOR variants (dashed lines, with colors representing different similarity metrics) using the same settings as in Sec.N. This result demonstrates that average running time for various variants of GMValuator on a single generated image, with the recalled number set to 200, is dominated by the re-ranking process. As a result, the running time is nearly independent of the dataset size. Noting that the number of recalled images during the re-rank phase is fixed, thus time cost reveal minimal variation in runtime with increasing dataset size, demonstrating that our method maintains consistent computational efficiency even as the data scale grows.

In Fig.11, we present the time cost for one generated image during the recall phase (No-Rerank), with dataset sizes ranging from 100 K to 50000 K. The results show that the running time remains small across different dataset size. Benefiting from the efficiency of our recall phase, the time overhead increases only marginally. In contrast, as noted in Kong & Chaudhuri (2021), "the run-time complexity of VAE-TracIn is linear with respect to the number of samples, checkpoints, and network parameters". Similarly, IF4GAN Terashita et al. (2021) is time-intensive, with complexity increasing alongside model size and the number of training epochs, as outlined in Algorithm 2 of their work.

# P  IMPACT OF QUANTIZATION RECALL

In this section, we evaluate the effect of quantization recall by empirically compare quantization recall with nearest neighbor (NN) search using varying recalled number $K$ during the recall phase. To be specific, We conducted our experiments on the MNIST dataset using a GAN model with recalled number $K$ increasing from 100 to 700. For consistency, we leverage $l_2$-distance in the reranking phase.

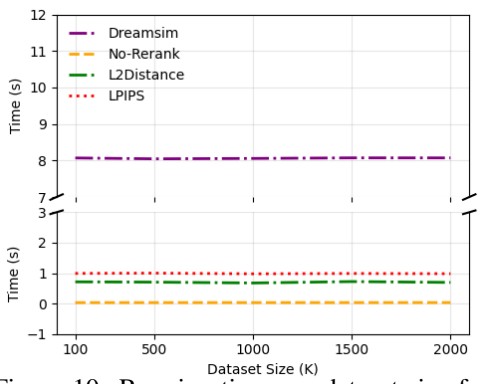

Figure 10: Running time vs. dataset size for variants of GMValuator.

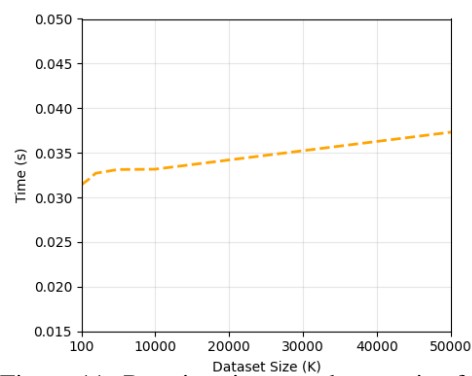

Figure 11: Running time vs. dataset size for recall phase (No-Rerank) of GMValuator.

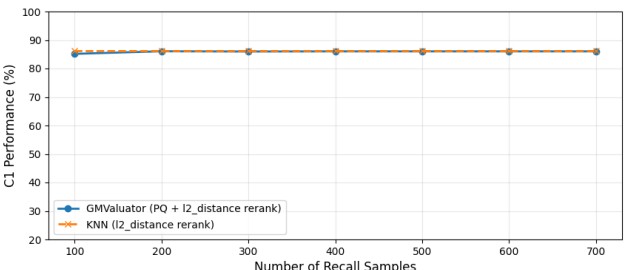

Figure 12: Running Time vs. Number of Recall Samples

***Efficiency.*** GMVALUATOR significantly outperforms nearest neighbor search in speed, taking an average of 0.25 seconds to attribute the most significant contributors for a generated sample, which is more than 50 times faster compared to 13.75 seconds for NN search. Whats more, the time cost for NN search will increase linearly with dataset size, while using PQ recall shows only a slight increased time in larger dataset size as shown in Fig 11. The experiments in this section are run in an environment equipped with an RTX A6000 GPU (48GB) and a 12-vCPU Intel(R) Xeon(R) Platinum 8255C CPU @ 2.50GHz.

***Effective.*** As illustrated in Fig. 12, under different recalled numbers (from 100 to 700), the results of GMValuator are similar to the exhaustive KNN search, demonstrating that our method maintains high performance while ensuring computational efficiency.

