# OpenReview forum: "GMValuator: Similarity-based Data Valuation for Generative Models"
_ICLR.cc/2025/Conference — ICLR 2025 Poster_

### Official Review · Reviewer_LEU4 · 2024-10-27

**Soundness:** 3
**Presentation:** 3
**Contribution:** 3
**Rating:** 6
**Confidence:** 3

**Summary:**

In this paper, the authors introduce Generative Model Valuator (GMVALUATOR), the first training-free and model-agnostic approach to providing data valuation for generation tasks. The authors formulate data valuation for generative models as an efficient similarity-matching problem. The paper further eliminates the biased contribution measurement by introducing image quality assessment for calibration. Also, the paper introduced four evaluation criteria for assessing data valuation methods in generative models.

**Strengths:**

The strengths of this paper are listed as the following:

1. GMVALUATOR is claimed to be the first modal-agnostic and retraining-free data valuation method for generative models.

2. The authors formulate data valuation for generative models as an efficient similarity-matching
 problem. The paper further eliminates the biased contribution measurement by introducing image
 quality assessment for calibration.

 3. The authors propose four evaluation methods to assess the truthfulness of data valuation and evaluate GMVALUATOR on different datasets, such as benchmark datasets and high-resolution
large-scale datasets, and various deep generative models to verify GMVALUATOR’s validity.

**Weaknesses:**

The weaknesses of the paper are listed as follows:

1. The paper only mentioned 4 criteria for assessing data valuation methods. They are: C1: Identical Class Test; C2: identical attributes test; C3: Out of Distribution Detection; and C4: Efficiency.

How about other criteria?
Why only use these 4?

For example, how about Cost-Benefit Analysis (i.e., the trade-offs between the costs of acquiring or processing data versus the performance gains from using it in model training)?
Please give some examples and formulas to measure the costs of data acquisition/processing. Then compare them with the performance gains of those data in the context of generative model training. This will be more practical for the proposed new approach.

2. The paper shall discuss the important aspects of data evaluation, such as accuracy and complexity. For example, how accurate is it for the proposed framework? How much is the complexity while implementing the proposed approach? Also, the authors shall discuss the proposed framework on accurately capturing the contributions of individual data points in various different scenarios.

3. The paper shall illustrate other aspects such as scalability, whether the proposed approach is useful for handling big data in real-time applications.  For example, the authors can provide the results of testing the proposed method on progressively larger datasets or measuring processing times for different data sizes.

**Questions:**

GMVALUATOR is claimed as the first modal-agnostic and retraining-free data valuation method for generative models. How about the comparison with other approaches such as Information-Theoretic Measures (Akhilan Boopathy et al., ICML 2023, "Model-agnostic Measure of Generalization Difficulty"), etc.?
Please illustrate whether other approaches are applicable to generative models in a model-agnostic and retraining-free manner, which is important to your claim.

---

> ### Author Response · Authors · 2024-11-23
> **Response Part 1**
>
> Thank you for your time and highlighting our contribution on model-agnostic, training-free design, quality score and extensive experiments. Please see our responses below. We hope they address your concerns!
>
> --------------_**W1& W2: Why use these 4 metrics? Shall also discuss the accuracy and complexity.**_--------------
>
> **Answer**: Thank you for your question.  First, we appreciate your suggestion on discussing accuracy and complexity and we agree these are important perspectives. We want to start by clarifying the lack of direct ground truth for data valuation. Therefore, we have used C1 and C2 as proxies to validate the accuracy. We also have discussed the complexity and using efficiency (C4) as measurement. Second, to address your insightful question on “why”, we appreciate the opportunity to clarify that our metrics are thoughtfully designed based on the principles of accuracy, efficiency, plausibility, and alignment with our theoretical framework to comprehensively evaluate data valuation methods for generative models. Below, we provide detailed clarification:
> - _C1 (Identical Class Test):_ This metric evaluates whether the most significant contributors from the training data belong to the same class as the generated data. By analyzing class-level consistency, it directly measures the alignment between data valuation and generative outputs. Notably, this metric is also used in VAE-TracIn, further validating its effectiveness. As shown in Table 2, GMVALUATOR achieves high accuracy with >88% class alignment for datasets like MNIST, highlighting the precision.
>
> - _C2 (Identical Attributes Test):_ Supported by Theorem 2.4, which establishes a bounded classification error for describable attributes, this metric examines attribute-level consistency between training and generated data. It captures nuanced relationships, such as features like gender or accessories. As presented in Table 3, GMVALUATOR achieves >95% attribute alignment, showcasing its ability to capture fine-grained relationships effectively.
>
> - _C3 (Out-of-Distribution Detection):_  Designed for plausibility, this metric evaluates the method’s ability to identify noisy or irrelevant training data. By ranking noisy samples, the results in Figure 5 show that GMVALUATOR assigns lower values to noisy data compared to IF4GAN, demonstrating its robustness in filtering irrelevant samples.
>
> - _C4 (Efficiency):_ This metric is designed to improve the computational efficiency of the data valuation method. As shown in Table 4, we measure the average time taken to attribute the most significant contributors for a single generated sample. GMVALUATOR demonstrates exceptional efficiency, being approximately 30 times faster than VAE-TracIn and 3 times faster than IF4GAN.
>
> These metrics collectively validate the accuracy, plausibility, and efficiency of GMVALUATOR, ensuring its reliability and practicality for generative model applications.

---

> ### Author Response · Authors · 2024-11-23
> **Response Part 2**
>
> -------------_**W1 Cost-benefit analysis**_------------
>
> **Answer:**
> Thank you for your comments. Conducting a cost-benefit analysis is indeed a valuable suggestion. However, as mentioned in lines 50–52 of our original submission (lines 50-56 in the revision), generative models currently lack robust performance metrics to accurately evaluate the performance of generative models (namely the benefit). Setting aside this limitation, we show indeed one can use such analysis to some extent. Specifically, here, we used Fréchet Inception Distance (FID) to assess performance gains using high-value data vs low-value data determined by GMValuator. For data acquisition/processing costs, we assumed a linear relationship proportional to the number of data points. The process is outlined as follows:
>
> - We fine-tuned a CIFAR-10 pre-trained model on the MNIST dataset and obtained the full dataset finetuned $\text{model}_f$.
>
> - We then calculated the score of each training data point given a set of generated data, identifying the top 1,000 high-value data points (Dataset H) and 1,000 low-value data points (Dataset L).
>
> - Then the pre-trained model was fine-tuned separately using Dataset H and Dataset L, resulting in $\text{model}_H$ and $\text{model}_L$, respectively.
>
> - We evaluate the performance gain by calculating the FID between the datasets generated by $\text{model}_f$ and those generated by $\text{model}_H$ and $\text{model}_L$ respectively.
>
> | Model         | FID |
> |--------------------------|-------------------------------------------|
> | $model_L$ (Low Value)  | 201.25                                                |
> | $model_H$ (High Value) | **139.52**                                           |
>
> As shown in the table, the significantly lower FID score for model_H (trained with high-value data) indicates that its distribution aligns more closely with the one generated by $\text{model}_f$. This demonstrates that given the same cost, using high-value data determined by GMValuator can achieve a larger gain with respect to FID.
>
> Although the above analysis supports our finding of high-value data, there are several vulnerable factors that remain concerns in the approaches:
>
> First, the generative models lack robust performance metrics to accurately measure benefits (as mentioned in lines 50–52 in our original submission or lines 50-56) and FID is only one of the possible choices while other benefit metrics for generative model performance may result in different conclusions which may lead to imprecise performance gain evaluations and, consequently, less reliable cost-benefit analyses.
>
> Second, we leveraged a pre-trained model for fine-tuning, as training a generative model from scratch requires vast amounts of data. This approach may be infeasible or unreliable for measuring performance gain when the total amount of training data (and total cost) is small. While introducing a pre-trained model can resolve this issue practically, it may also introduce biases, as different choices of pre-trained models can affect the cost-benefit results.
>
> —-----------_**W2  Should test in various scenarios**_—-----------
>
> **Answer:**
> Thank you for your question. GMVALUATOR has been thoroughly tested across a wide range of datasets and models in our original submission, as detailed below:
>
> We conduct evaluation on different scale of datasets:
>  - _Small-scale datasets (e.g., MNIST and CIFAR-10):_ Validates its effectiveness in simpler, low-dimensional scenarios.
>  - _High-resolution datasets (e.g., CelebA, FFHQ, and AFHQ):_ Demonstrates its capability to handle complex, high-dimensional data.
>  - _Large-scale datasets (e.g., ImageNet):_ Highlights its scalability and robustness in real-world, large-scale applications.
>
> We have also examined different types of generative models in our experiments (i.e., GAN, VAE and Diffusion model), listing them in Table 10 of the appendix. Additionally, we included text-image diffusion models in Table 14 of the revised version.
>
> Our comprehensive experiments demonstrate GMvaluator’s adaptability and robustness, as also acknowledged by reviewers eyah and 35uE.

---

> ### Author Response · Authors · 2024-11-23
> **Response Part 3**
>
> —----------_**W3: Scalability on progressively larger dataset**_—-----------------
>
> **Answer:**
> Thank you for your suggestions. Based on your feedback, we conducted additional experiments on progressively larger datasets.
>
> In Fig. 10 of the revised manuscript, we show the total running time for one generated image with dataset sizes ranging from 100 K to 50000 K., noting that the number of recalled images during the re-rank phase is fixed at 200. The results reveal minimal variation in runtime with increasing dataset size, demonstrating that our method maintains consistent computational efficiency even as the data scale grows. This underscores the scalability of GMValuator and highlights its suitability for large-scale applications without significant overhead.
>
> Specifically, in Fig. 11 of the revised manuscript, we present the average time cost for recalling contributors for one generated image during the recall phase. Benefiting from the efficiency of our recall phase, the time overhead increases only marginally. In contrast, as noted in VAE-TracIn, "the run-time complexity of VAE-TracIn is linear with respect to the number of samples, checkpoints, and network parameters". Similarly, IF4GAN is time-intensive, with complexity increasing alongside model size and the number of training epochs, as outlined in Algorithm 2 of their work.
>
> —---------------_**Q1: First modal-agnostic and retraining-free method**_—------------------
>
> **Answer:**
> Thank you for your question. We are indeed the first modal-agnostic and retraining-free data valuation method for generative models.  Following your suggestion, we compared [1] and summarized the key points to support our claim.
>
> - _[1] requires retraining but ours does NOT_: [1] is not training-free, it requires sampling all hypotheses from a broad and general hypothesis space encompassing all model classes practically applicable to a task.  And it then measures the hypotheses' loss on the testing data and training data and calculates the difficulty of finding well-generalizing models among the ones that fit the training data.  Therefore, to our best knowledge, we still believe that our statement as “the first modal-agnostic and retraining-free method” holds.
>
> - _[1] is a metric-based method but ours is NOT_: As shown in definition 1 in [1], It requires calculating loss thus falling at the same methodology with metric-based methods (accuracy or loss) like VAE-TracIn and IF4GAN to evaluate the value, as state in lines 49-52 in our original submission or lines 50-55 in the revision, arising from lacking robust performance metrics in generative models.
>
> - _[1] Focuses on data trends but ours can provide explicit data values_:  As stated in the abstract of [1], they aim at giving a trend of a set of training data rather than value for a single data point. Also, in section G.3 in appendix, the author observes that changing the amount of training data, even by many orders of magnitude, does not significantly affect the amount of inductive bias required to generalize due to the absence of strong constraints on the hypothesis space in achieving model agnostic. This limits its applicability to evaluate the value of datapoint.
>
> We would like to emphasize again that our work focuses on the need for training-free and model agnostic data valuation methods for image generative models. We also have denoted the limitations of using metric-based methods. Although [1] is an interesting paper, we don't believe it is applicable to provide data valuation under our focused setting due to its dependency on performance metrics, training requirements, focus on dataset-level trends, and scalability issues, GMVALUATOR uniquely addresses these challenges by leveraging a training-free model-agnostic similarity-matching framework, making it a pioneering contribution in this area. Following your valuable suggestion, we have discussed [1] in related work in Appendix C.1 in our revision.
>
> [1] Model-agnostic Measure of Generalization Difficulty

---

### Official Review · Reviewer_35uE · 2024-11-04

**Soundness:** 4
**Presentation:** 3
**Contribution:** 3
**Rating:** 8
**Confidence:** 4

**Summary:**

This paper proposes a novel data valuation method for generative models. The authors introduce a model-agnostic, training-free data valuation framework, addressing a significant challenge in the field: existing methods typically require retraining or Hessian calculations, which are computationally intractable. In this approach, the contribution of a training data point to a generated data point is defined as inversely proportional to the distance between the two data points. The author performed extensive experiments against two baselines and showed effectivenss of the propsoed method.

**Strengths:**

- The paper introduces a novel and intuitive idea for data valuation in generative models, and the results are promising.
- The experiments are well-designed, exploring multiple distance functions and encoders to validate the approach. Also, multiple test scenarios were covered, all showing good supporting results for the proposed method.
- The paper is well-written and easy to follow, effectively conveying the methodology and findings.
- The paper covers relative literature well.

**Weaknesses:**

- The impact of the quantization step on the final results is not explored. Understanding this effect would provide a clearer picture of the method’s performance.
- While section 2 introduces some underlying assumptions and a theoretical motivation for using a similarity-guided data valuation score (illustrated in Figure 1), the framework would benefit from a more rigorous theoretical foundation. Further studies on theoretical support could strengthen the framework’s conceptual grounding and its reliability across different applications.

**Questions:**

- What is the effect of the quantization step in the recall phase?
- Empirically, how different is the proposed score from a nearest neighbor search with the rerank metric in the embedding space?
- How well does the proposed method generalize to other generative models, such as diffusion models?

---

> ### Author Response · Authors · 2024-11-23
>
> Thank you for your time and for highlighting our method's innovation, extensive experiments, promising results and clear writing. Please see our responses to your questions below. We hope these address your concerns!
>
> --------_**W1 & Q1,2 : Impact of quantization and comparison to NN search.**_----------
>
> **Answer:**
> Thank you for your insightful question. The impact of the quantization step is to improve the efficiency in the recall phase as stated in lines 286-287 of our original submission (293-294 in revision). This approach avoids the significant computational overhead of exhaustive nearest neighbor search by using a quantized retrieval step in the recall phase to narrow down candidates, followed by a precise reranking phase applied only to this smaller subset.
>
> Following your valuable suggestions, we have conducted an experiment on the MNIST dataset using a GAN model to demonstrate the effectiveness of the quantization step:
>
> - _More Efficient_: PQ recall strategy in our method significantly outperforms nearest neighbor (NN) search in speed, taking an average of 0.25 seconds to attribute the most significant contributors for a generated sample, which is more than **50 times faster** compared to 13.75 seconds for NN search. What's more, the time cost for NN search will increase linearly with dataset size, while using PQ recall shows only a slight increased time in larger dataset size as shown in Fig 11 in revision.
>
> - _Effectiveness:_ we empirically evaluated the performance of varying the recalled number K during the recall phase. As shown in the Fig 12 in revision, under different recalled numbers ( from 100 to 700), the results of GMValuator are nearly the same as the exhaustive NN search, demonstrating that our method maintains high performance while ensuring computational efficiency.
>
> The above advantages demonstrate the scalability of our approach and makes it more suitable for large-scale real-world applications, where exhaustive KNN search becomes impractical.
>
> ----------_**W2: More rigorous theory**_-----------
>
> **Answer:**
> Thank you for the suggestions. We would like to take this chance to clarify that our theory is carefully derived, as demonstrated through the following breakdown of our proof sketch:
>
> - _Motivation (Section 2.1):_  We observed that an optimal generator closely approximates the training distribution.
>
> - _Attribute-Based Characterization (Definition 2.1):_  Images are described through discrete attributes, a widely supported approach in prior work [1,2,3] (e.g., describing a dog's appearance). By employing an attribute labeling function \( f \), we map both training and generated data into a shared attribute space, ensuring that the data valuation score aligns with interpretable and semantically meaningful characteristics.
>
> - _Definition of Contributors (Definition 2.2):_  We provide a precise definition of contributors in generative models, establishing a robust theoretical framework for identifying subsets of training data that significantly impact generated samples.
>
> - _Practical Applicability via Bounded Classification Error: (Theorem)_ To ensure practical feasibility, we incorporate bounded attribute classification error, allowing approximate contributors to be identified by selecting those closest to the generated samples.
>
> Our empirical results strongly support this theoretical foundation. The T-SNE plot in Figure 1 and the Identical Attributes Test in Section 4.4 confirm alignment between theory and practice, demonstrating GMVALUATOR's capacity to reliably identify key contributors closely aligned with generated data.
>
> We are open to further discussions and aim to refine our theoretical framework with additional insights in future work.
>
> [1] Describing Objects by Their Attributes
> [2] Learning Discrete Concepts in Latent Hierarchical Models
> [3] Attributes of Images in Describing Tasks
>
> ----------_**Q3: Other generative models, such as diffusion models**_------------
>
> **Answer:**
> Thanks for the feedback. In the experiments, we indeed have examined different kinds of diffusion models, including Diffusion-StyleGAN,Class-free Guidance Diffusion, and other diffusion models. The types of diffusion models in our experiments are listed in Table 10 of the appendix and the results of using these diffusion models are shown in Table 3 (Diffusion-StyleGAN), Table 7 (Class-free Guidance Diffussion), Table 8 (Diffusion-StyleGAN), Figure 4 (Diffusion-StyleGAN), FIgure 7 (Diffusion, Class-free Guidance Diffusion), etc.

---

> > ### Comment · Reviewer_35uE · 2024-11-25
> > **Thanks for the reply**
> >
> > I thank the authors for carefully addressing my questions. I believe this work has great impact especially with the wide application of generative models now and will maintain my score.

---

> > > ### Author Response · Authors · 2024-11-25
> > > **Thank you**
> > >
> > > Thank you for your thoughtful feedback and continued support. We’re pleased that our response addressed your concerns and grateful for your recognition of our work’s great impact. Your encouragement means a lot to us—thank you!

---

### Official Review · Reviewer_smoK · 2024-11-04

**Soundness:** 4
**Presentation:** 4
**Contribution:** 3
**Rating:** 6
**Confidence:** 3

**Summary:**

The paper focuses on the data valuation of generative models. Existing data valuation methods designed for discriminative models cannot adapt to generative models due to 1) lack of robust performance metrics; 2) the large size of generative models; and 3) lack of data labels. In order to mitigate this gap, the authors propose GMValuator. GMValuator is based on similarity matching between training data and generated data. If a training sample is similar to a generated sample, it is considered to have contributed to the generated sample. The value of a training sample is computed by the quality of its contributed generations. Four evaluation criteria are introduced to assess data valuation. Experiments demonstrate the effectiveness of the proposed GMValuator.

**Strengths:**

- This is the first paper on data valuation on generative models. Previous data valuation methods focus on discriminative models and cannot adapt to generative models.
- Compared to the retraining-based and influence-based methods, GMValuator is efficient. It does not require any retraining or computation of hessian.
- GMValuator is effective on the proposed metrics. GMValuator has significantly improved compared to baseline methods.

**Weaknesses:**

- For SOTA text-to-image models like stable diffusion, the image domain is much wider than the test models. As a result, a large number of generated images may be required for accurate data valuation. Meanwhile, generation with these models is slow. More results and ablation on stable diffusion on the SOTA text-to-image models would be helpful.
- While the proposed metrics are intuitively reasonable, it is coarse-grained and may not be able to reflect the effectiveness of data evaluation methods.

The authors addressed my concerns.
I raise my score to 6.

**Questions:**

Please refer ti weaknesses.

---

> ### Author Response · Authors · 2024-11-23
> **Response to Weakness 1**
>
> Thank you for your time and for highlighting our method's innovation, efficiency, and effectiveness. Please see our responses below to your questions. We hope these address your concerns. We are happy to address any additional questions that you may have.
>
> ---------_**W1:Clarification on No needs for large generated data**_-----------
>
> **Answer:** There appears to be a misunderstanding regarding the requirements of our proposed method. As defined in our paper, data valuation refers to the contribution of training data to a given set of generated data (lines 79-82 in the original submission, lines 82-85 in the revision). This generated data can be a small subset and may represent a subdomain of the training image domain (lines 129–130 in original submission, lines 134-136 in revision). Also, our theory does not assume that the generated data must cover a diverse distribution. As shown in our experiment in Figure 3, when the generated data belongs to a subdomain (e.g., dog images), the training images within the same subdomain (e.g., dogs in the training data) are assigned higher values than other data. For additional results, please refer to C1 (Table 2) and C2 (Table 3). These results also demonstrate that the value of training data can vary depending on the given generated data being different.
>
>
> -----------_**W1: Results on stable diffusion.**_---------
>
> **Answer:** Following your suggestion, we conducted additional ablation experiments using Stable Diffusion fine-tuned with LoRA on a Naruto dataset. The results, presented in Table 14 in the revised manuscript, demonstrate that our approach remains effective in evaluating data contributions.
>
> Regarding computational efficiency, we report the time required to identify significant contributors to a single generated sample in Table 15 in the revised manuscript. Our method’s efficiency demonstrates the practical feasibility of our approach with test-image models like Stable Diffusion.
>
> Thanks again for the suggestion. We have incorporated these new results into the Appendix O of our revision.

---

> ### Author Response · Authors · 2024-11-23
> **Response to Weakness 2**
>
> ---------_**W2: Evaluation metric.**_-----------
>
> **Answer:**
> We appreciate your comment and the opportunity to address this point.
> We acknowledge that evaluating the effectiveness of data valuation methods is inherently challenging due to the lack of an established ground truth or direct measurement standards.
> While the metrics may appear coarse-grained, we believe they are still effective in capturing the critical aspects of data valuation. Importantly, coarse-grained does not equate to ineffectiveness; rather, it reflects the practical constraints and the need to make meaningful comparisons in the absence of definitive ground truth.
>
> In light of this, we carefully designed our metrics to provide meaningful insights into the relative performance and utility of different methods within the scope of our study. Specifically, our metrics are thoughtfully designed by following our principles of efficiency, plausibility and our theorem to comprehensively evaluate the effectiveness of data valuation methods in generative models. Below, we provide further clarification:
>
> - _C1 (Identical Class Test):_ Notably, this metric follows the employment in existing work VAE-TracIn (Kong & Chaudhuri), further validating its effectiveness. This metric determines whether the most significant contributors among the training data belong to the same class as the generated data. It provides a fine-grained assessment by analyzing the class-level consistency of individual training samples with specific generated outputs. This directly highlights the alignment between data valuation and generative outputs.
>
> - _C2 (Identical Attributes Test):_ Supported by Theorem 2.4, which establishes a bounded classification error for data with describable attributes, this metric extends the evaluation to attribute-level granularity. By comparing the consistency of specific features (e.g., gender, accessories) between training and generated data, it captures nuanced relationships. The results show GMVALUATOR effectively identifies key contributors closely matching the generated samples, showcasing its practical utility and superior performance.
>
> - _C3 (Out-of-Distribution Detection):_ This metric designed for plausibility assesses whether the data valuation method can effectively identify noisy or irrelevant training data. By analyzing the ranking of noisy samples, the metric provides a detailed evaluation of the robustness of the valuation method in distinguishing high-contribution and low-contribution data points.
>
> - _C4 (Efficiency):_ This metric designed for computational efficiency of the data valuation method. It quantifies the time requirements for the method, ensuring that it can be applied efficiently in real-world scenarios.
>
> - _Consistency Across Metrics:_ Another significant strength of our framework lies in the consistent and complementarity of results across all metrics—C1, C2, and C3. Together, these metrics provide a holistic validation of GMVALUATOR, confirming its effectiveness across multiple dimensions of evaluation.
>
> In conclusion, we are confident that these metrics form a robust and comprehensive foundation for evaluating data valuation methods. We are happy to address any additional questions that you may have.

---

> ### Author Response · Authors · 2024-11-26
> **Thank you**
>
> Thank you for taking the time to review our rebuttal and for your thoughtful feedback. We’re delighted that our response has addressed your concerns. We truly appreciate your engagement and your increased score. Your encouragement means a lot to us—thank you!

---

### Official Review · Reviewer_eyah · 2024-11-08

**Soundness:** 3
**Presentation:** 2
**Contribution:** 3
**Rating:** 6
**Confidence:** 3

**Summary:**

The paper introduces GMVALUATOR to solve the data valuation problem for vision generative models, which is important yet has been overlooked by existing studies. GMVALUATOR formulates data valuation as a similarity-matching problem and incorporates image quality assessment to calibrate the contributions of data samples. Despite high computational complexity with large datasets, GMVALUATOR demonstrates effectiveness across multiple datasets and generative models, positioning it as a promising tool for data valuation in the field. Although the technical contribution is obvious, there is still much room for improvement in the author's presentation and content arrangement.

**Strengths:**

1.	The authors proposed GMVALUATOR to tackle the data valuation issue for generative models. GMVALUATOR is innovative, and model-agnostic, enabling broad applicability and adaptability across various generative models. Besides,  GMVALUATOR does not require retraining of models, offering the advantage in R&D scenarios with limited computational resources.
2.	The authors provide detailed theoretical justification for formulating data valuation for generative models as a similarity-matching problem. They also provide empirical validation for this motivation. This makes the GMVALUATOR being very reasonable.
3.	The authors demonstrated through extensive experimental results that the data valuation method proposed can achieve good results in multiple aspects and over multiple generative models. Besides, they also provided validation of the experimental settings and the open-source implementation, which further increased the credibility of the results.

**Weaknesses:**

1.	The manuscript is not well written. For example, in the Introduction, before talking about the existing work, it is suggested to generally define/introduce the data valuation problem (including the input and the objective). Moreover, the authors didn't highlight the urgent need for data valuation in existing generative models; this poses a challenge to the motivation of this paper. Most importantly, instead of briefly introducing the principle of the proposed GMValuator (such as why and how to formulate data valuation for generative models as an efficient similarity-matching problem), the author only introduces the goal to be achieved. These places make the content difficult to read.
2.	Although the authors claim that they proposed a versatile data valuation method for generative models, in both the problem formulation, method introduction, and subsequent experiments, they only evaluate it on image samples. Therefore, the statement "for generative models" may be inappropriate. Even though the authors mentioned in the Current Limitation "However, this does not mean that GMVALUATOR cannot be easily adapted to Natural Language Processing (NLP) fields, given its core idea of similarity matching." However, the reason for this statement may need further explanation.

**Questions:**

1.	As stated by the authors: However, this does not mean that GMVALUATOR cannot be easily adapted to Natural Language Processing (NLP) fields, given its core idea of similarity matching. So,  how to apply the proposed method to generative models for text-based data or data from multiple modalities?
2.	If data valuation is not used in training generative models, what will be the limitations? Can the proposed method or evaluation metrics measure the limitations?

---

> ### Author Response · Authors · 2024-11-23
> **Response Part 1**
>
> Thank you for your time and highlighting our method's innovation, strong theoretical evidence and extensive experiments. Please see our responses below to your questions. We hope these address your concerns.
>
> ------_**W 1.1 Definition and highlighting urgent needs**_------
>
> **Answer**: Thank you for your valuable suggestions on writing.
>
> **Problem definition** We have mentioned the general definition of DV as “contribution measurement of each training sample” in line 38 in our original submission (line 38 in revision) and provided the problem definition of DV we focus on as “data valuation as the contribution to the given generated data” in line 79-80  in our original submission (line 83-84 in revision). We tried to make the introduction concise and leave a more detailed definition in Section 2. Following your suggestion, we added a footnote in line 35 in revision to give a clear point to the definition in Section 2.
>
> **Urgent needs**: We respectfully notice that we indeed have described the need for data valuation in generative models in our original submission, as outlined below:
>
> - _Most focus on discriminative models_ : In the second paragraph, we highlighted that existing data valuation studies primarily target supervised learning for discriminative models. Additionally, we detailed related work in this area to establish context.
>
> - _Urgent need on  generative models_ :
> In the third paragraph, we explicitly stated, "Data valuation in the context of generative models has **NOT** been well-investigated in the current literature", with "NOT" bolded to emphasize the urgency. We also identified key challenges, such as the absence of robust metrics in metric-based methods and the inefficiency of approaches reliant on labeled data.
> In the fourth paragraph, we analyzed existing generative model methods, noting their dependency on specific architectures (e.g., IF4GAN for GANs and VAE-TracIn for VAEs), which restricts generalizability.
>
> - _Our motivation and goal_: In the fifth paragraph, we addressed these urgent needs, outlining our goal to develop a model-agnostic, efficient, plausible, and truthful data valuation method.
>
> These points collectively articulate the necessity for data valuation in generative models and establish the motivation for our work.  Following your suggestion, we use “Our motivation and goal: ” (line 73 in revision) and “Urgent need on generative models” (line 47 in revision) to provide clear pointers to the content.
>
>
> --------_**W 1.2 Introducing the principle of GMValuator**_-----
>
> **Answer:** Thank you for your comment. In the introduction, we provided a brief overview of the principle behind GMVALUATOR in lines 82-83 in our original submission (lines 85-86 in revision) to maintain a balance between clarity and brevity. Our goal was to ensure accessibility for a broad audience while avoiding overwhelming readers with excessive detail upfront.
>
> We did, however, provide a comprehensive explanation of the rationale for formulating data valuation as a similarity-matching problem in Section 2, where we detailed the theoretical motivation and underlying principles. Additionally, the implementation details and methodology for achieving this are thoroughly explained in Section 3.
>
> Based on your valuable feedback, we have added a footnote at line 86 in the revision to explicitly guide readers to the sections discussing the "why" and "how" in greater depth.

---

> ### Author Response · Authors · 2024-11-23
> **Response Part 2**
>
> -----_**W2&Q1. Clarification on the statement in appendix**_-----
>
> **Answer:** We apologize for the confusion and would like to begin by clarifying that, in this paper, our focus was on evaluating GMValuator specifically with image-based generative models. For instance, we referenced image quality in line 21 in original submission and described images with attributes in Definition 2.1 of our original submission. To address your concern, we have further clarified this in our revised version. Additionally, we are open to incorporating any further suggestions from the reviewers to improve clarity and address this point comprehensively.
>
> Regarding our statement in the "Limitations and Future Directions" section (Appendix L), we clarified that the core idea of GMValuator—similarity matching—is not inherently tied to a specific data type or generative model, provided the assumptions in our theorem are satisfied. The phrase “does not mean GMValuator cannot be easily adapted” was intended to reflect the theoretical flexibility of our approach, rather than to assert that adaptation to other modalities is straightforward or immediate. While our current work focuses on image-based models, we hope it inspires future studies to explore suitable similarity matching strategies and formulations for text-based data, multimodal data, or other domains. To adapt to NLP, we envision that one needs to modify the following parts of our method: change image embedding to text embedding modules (such as BERT), find suitable similarity metrics in the rerank phase in Efficient Similarity Matching Module, and find suitable quality score measurement for text.
>
> To support such extensions, we will open-source our code, encouraging the community to adapt GMValuator to NLP and other fields. We are excited about the possibility of collaboration and further discussion to refine and expand our method across diverse generative tasks.

---

> ### Author Response · Authors · 2024-11-23
> **Response Part 3**
>
> -----_**Q2. Limitations of existing data Valuation methods that rely on training**_------
>
> **Answer:**  We answer the limitations of existing data valuation methods that rely on training generative models and show how our metrics measure these limitations below:
>
> - *Limited Performance:* As discussed in lines 50-52 in our original submission (50-54 in revision) , metric-based methods like IF4GAN and Trac-Inf lack robust performance metrics, leading to suboptimal results and ultimately hindering their overall effectiveness.
>
>    **Performance Metric:** Our experimental results (C1 and C3) were utilized to evaluate and confirm these performance limitations.
>
> - _Limited Efficiency:_ These methods are computationally expensive. For instance, IF4GAN's complexity scales significantly with increases in model size and the number of training epochs, as described in Algorithm 2 of their work. Similarly, VAE-TracIn involves Hessian estimation for influence function calculations, which results in high computational costs.
>  **Efficiency Metric:** C4 was used to measure the average valuation time for one generated sample to highlight the inefficiency of these methods.
>
> -  _Limited Scalability:_ These methods are not model-agnostic, making them difficult to generalize across different types of generative models. For example, IF4GAN is tailored to GANs, while VAE-TracIn is specific to VAEs.
> **Scalability Evaluation:** To address this limitation, we evaluated diverse generative models, including diffusion models, VAEs, and GANs. These experiments demonstrate the model-specific constraints of IF4GAN and VAE-TracIn, as they cannot be applied beyond their respective model types.
>
> This comprehensive analysis using performance (C1, C3), efficiency (C4), and scalability evaluations underscores the limitations of existing methods and highlights the adaptability and robustness of our proposed approach.

---

### Comment · Reviewer_LEU4 · 2024-11-25

Thanks for the replies and further illustrations. Please continue to polish the paper. It is hard to make big changes now.

I have given the maximum possible high score. Hence, I will keep the score.

---

> ### Author Response · Authors · 2024-11-25
> **Reasons for no big changes**
>
> Thank you for your feedback and the thoughtful discussion of our revisions. We appreciate the acknowledgment of our responses and additional experiments, as we have committed our best efforts to satisfy the reviewers' requests. However, this does not imply that we plan to make big changes. We detail the reasons below, which we hope will effectively address your concerns:
>
> - **W1 & W2 are already sufficiently detailed in Section 4.2**: The contributions of W1 and W2 were explicitly detailed in Section 4.2 of the original manuscript. We believe that the current presentation comprehensively encapsulates the significance of these metrics, and therefore, further changes would not substantively enhance the paper.
>
> - **Cost-benefit analysis is an interested suggestion but is deliberately omitted**: In our original submission, we outlined the limitations of involving performance metrics (lines 50-52 of the original manuscript). While we provided additional results on the cost-benefit analysis in the rebuttal, we also discussed the potential issues and decided **not** to integrate them into the main text.
>
> - **Remaining the same statement for our contribution and novelty (Q1)**: We maintain that our approach is the first modal-agnostic and retraining-free method. Although [1] may appear relevant, it is not specifically tailored for data valuation, and we have thoroughly discussed its limitations. Consequently, we do not find it necessary to integrate it into the main paper, although we have included a discussion in the appendix.
>
> - **Add W3 (Scalability) to appendix**: We thank the reviewer for their question on scalability and have added an extra experiment in the appendix. However, as highlighted in our response to W2, we have **already** demonstrated our method's scalability in the original submission: we evaluated datasets ranging from small to large and examined various generative models in our experiments (GAN, VAE, and diffusion models).
>
> Once again, thank you for your time and constructive engagement with our work.

---

### Meta-Review · Area_Chair_Vks7 · 2024-12-17

**Metareview:**

This paper studies data valuation for generative models which is a topic not much explored in the research community. It proposes a training-free and model-agnostic approach based on similarity matching. It also introduces four evaluation criteria.

Major strengths:
- Data valuation for generative models is an important topic that has not yet received sufficient attention.
- The proposed method has the advantages of being training-free, model-agnostic, and efficient.
- Theoretical justification is provided for formulating the data valuation problem as similarity matching.
- Extensive experiments are conducted to support the claims.

Major weaknesses:
- Experimental validation is only conducted on image data but not other data modalities.

We thank the authors for addressing the comments raised by the reviewers in their reviews and revising their manuscript to address some of them.

**Additional Comments On Reviewer Discussion:**

The overall rating was increased during the discussion period.

---

### Decision · Program_Chairs · 2025-01-22

Accept (Poster)